# Ultrafast light targeting for high-throughput precise control of neuronal networks

Giulia Faini [1,2], Dimitrii Tanese [1,2], Clément Molinier[1], Cécile Telliez[1], Massilia Hamdani[1], Francois Blot[1], Christophe Tourain[1], Vincent de Sars [1], Filippo Del Bene [1], Benoît C. Forget[1], Emiliano Ronzitti [1] ✉ & Valentina Emiliani [1] ✉

Two-photon, single-cell resolution optogenetics based on holographic light-targeting approaches enables the generation of precise spatiotemporal neuronal activity patterns and thus a broad range of experimental applications, such as high throughput connectivity mapping and probing neural codes for perception. Yet, current holographic approaches limit the resolution for tuning the relative spiking time of distinct cells to a few milliseconds, and the achievable number of targets to 100-200, depending on the working depth. To overcome these limitations and expand the capabilities of single-cell optogenetics, we introduce an ultra-fast sequential light targeting (FLiT) optical configuration based on the rapid switching of a temporally focused beam between holograms at kHz rates. We used FLiT to demonstrate two illumination protocols, termed hybrid- and cyclic-illumination, and achieve sub-millisecond control of sequential neuronal activation and high throughput multicell illumination in vitro (mouse organotypic and acute brain slices) and in vivo (zebrafish larvae and mice), while minimizing light-induced thermal rise. These approaches will be important for experiments that require rapid and precise cell stimulation with defined spatio-temporal activity patterns and optical control of large neuronal ensembles.

Optogenetic neuronal excitation using single-photon widefield illumination has already proven its enormous potential in neuroscience, enabling the optical manipulation of entire neuronal networks with cell-type specificity and to disentangle the role of specific cell types in the control of specific behaviors[1,2]. However, establishing how a defined spatiotemporal activity pattern impacts a particular behavior, or how functionally identical neurons are connected and involved in a particular task, requires the precise control of single or multiple cells independently in space and time. This is now possible because of joint progress in opsin engineering, light-shaping approaches and high-power fiber laser development. Today, using two-photon (2P) holographic illumination of fast-photocycle-, soma-targeted-opsins, permits single or multi-spike generation with cellular resolution, sub-

millisecond precision and high spiking rates deep in tissue[3]. Using multiplexed spiral scanning[4] or multiplexed temporally focused light-shaping approaches[5–7], combined with high-energy fiber lasers and soma-targeted opsins[8,9], also enables simultaneous control of multiple targets in ~mm³ volumes at cellular resolution[10–12].

The unprecedented spatiotemporal precision reachable with the combination of these approaches, which we termed circuit optogenetics[3], has enabled high-throughput connectivity mapping in mouse cortex[13,14] or living zebrafish larvae[15] and probing inter-layer functional connectivity in mouse retina[16]. Combined with two-photon Ca²⁺ imaging and behavioral assays, circuit optogenetics has been used to show that the activation of a few cells can bias behavior by triggering the activity of precisely defined ensembles in the mouse cortex[11,12,17,18]

[1]Sorbonne Université, INSERM, CNRS, Institut de la Vision, F-75012 Paris, France. [2]These authors contributed equally: Giulia Faini, Dimitrii Tanese. ✉e-mail: emiliano.ronzitti@inserm.fr; valentina.emiliani@inserm.fr

or that selective stimulation of a small number of hippocampal place cells is sufficient to bias mice behavior during a spatial memory task[19]. Importantly, sequential projection of multiple holographic patterns at variable time intervals in the mouse olfactory bulb has revealed how the perceptual responses of mice depend on both the specific group of cells and cell numbers activated and on their relative activation latency[20].

These pioneering works suggest several exciting experimental paradigms for circuit optogenetics, for instance, the investigation of the temporal bounds of functional connectivity within which neurons "fire and wire together", or how many targets need to be activated to perturb complex behavioral responses or how large neuronal ensembles, spanning across multiple cortical layers, are functionally connected. Answering these questions requires the capability to manipulate neuronal activity at fine (sub-millisecond) temporal scales and/or large cell populations, which ultimately requires overcoming the current intrinsic technological limitations of holographic light patterning, specifically the limited temporal resolution for sequential control of neuronal activity and the high illumination power necessary for multitarget excitation.

Multitarget excitation uses holographic light shaping to multiplex the excitation beam to multiple locations, combined with either spiral scanning or multiplexed temporally focused light shaping (MTF-LS) approaches[21,22]. In spiral scanning approaches, a LC-SLM is used to multiplex the illumination beam into several diffraction-limited spots which are scanned in spiral trajectories using a pair of galvanometric mirrors (GMs). Several MTF-LS approaches have been developed[5], which differ in the approach used for light patterning. Generally, they are comprised of three units: (1) a beam shaping unit which sculpts light into particular forms, (2) a diffraction grating placed in a conjugate image plane to confine photostimulation to an axial region with cellular dimensions, and (3) a LC-SLM to multiplex

the sculpted light to multiple sample locations (Fig. 1a). This configuration admits multiple variants depending on the beam shaping unit used, which also defines the extent of the beam profile at the multiplexing LC-SLM[5]. Beam shaping units based on computer-generated holography illuminate the second LC-SLM with either a single chirped hologram of the size of the LC-SLM matrix[6] or with multiple chirped holograms[23]. Conversely, the use of an expanded Gaussian beam[7,24] or of the generalized phase-contrast method[6], produces a chirped horizontal line-beam spanning one dimension of the LC-SLM[7] (Supplementary Fig. 1).

In all described approaches, sequential generation of independent illumination patterns is achieved by projecting multiple holograms at a rate limited by the LC-SLM refresh rate (60–300 Hz) and cell illumination times (from ~ms to tens of ms). Moreover, the scattering, the available laser power and/or the necessity of maintaining brain temperature within physiological thresholds[10,25,26], when using powerful fiber lasers[3,21,27] have limited so far the number of targets simultaneously activable to 50–200 cells.

To overcome these limitations, we present an optical approach for ultrafast sequential light targeting (FLiT) based on the rapid displacement of temporally focused sculpted light through multiple, vertically aligned, holograms. Using FLiT, we introduce two illumination methods for multitarget optogenetics, termed *hybrid-* and *cyclic-* illumination, and demonstrate sub-millisecond control of sequential neuronal activation and low-power parallel multicell illumination, respectively.

*Hybrid*-FLiT illumination may enable arbitrary desynchronization (or synchronization) of neuronal ensembles and facilitate the study of the influence of spike timing on synaptic integration, plasticity, and information coding. *Cyclic*-FLiT illumination permits to considerably reduce the necessary power for multitarget excitation, thus keeping multitarget excitation below the threshold for thermal damage and

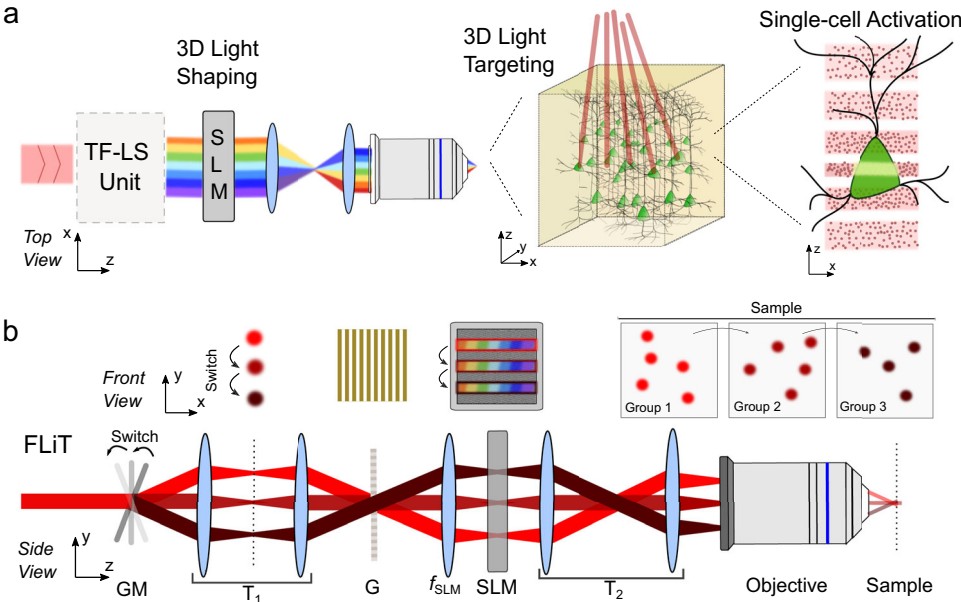

**Fig. 1 | FLiT optical scheme. a** General optical scheme for temporally focused light shaping. A temporally focused light-shaping architecture (TF-LS) allows (i) sculpting light into specific patterns and (ii) temporally focusing the photons to confine photostimulation to a shallow axial region with cellular dimensions. A subsequent LC-SLM modulation allows multiplexing the sculpted light to multiple 3D sample locations (dots in the beam represent photons). **b** Optical setup of FLiT. A pulsed collimated beam (red line) is reflected by a galvanometric mirror (GM) onto a diffracting grating (G) via a *4f*-telescope (T₁). Diffracted off the grating, the beam is projected onto a liquid-crystal spatial light modulator (SLM) by a f_SLM lens in the

form of a horizontal (i.e., orthogonal to the orientation of the grating lines) spatially chirped strip of light. The LC-SLM is projected onto the back aperture objective (OBA) via a telescope (T₂) so that ad hoc phase modulation on the LC-SLM allows multiplexing the initial beam and generating a multi-site temporally focused pattern of light in the sample. As deflection of the beam by the GM results into a translation of the illuminated bands on the LC-SLM (dark-red lines), addressing the LC-SLM with H independent tiled holograms φᵢ can lead to fast switch of different groups of light patterns into the sample. The top and bottom drawing represents the *XY* and the *YZ* plane views, respectively.

enables to scale up the number of simultaneously activable cells for multitarget optogenetics.

## Results

### Ultrafast light targeting (FLiT)

Here we introduce an optical configuration for ultrafast sequential light targeting (FLiT), where the multiplexing LC-SLM is addressed with multiple vertically tiled holograms and a galvanometric mirror (GM) is incorporated upstream to sweep the chirped expanded gaussian beam between holograms which generate sequential 2D or 3D illumination patterns (Fig. 1 and Supplementary Movies 1 and 2). We use this configuration to demonstrate two illumination paradigms for 2P multitarget optogenetics: *hybrid*-illumination, which achieves submillisecond control of sequential neuronal activation and *cyclic*-illumination, which gives high-throughput parallel multicell illumination, and low-temperature increase.

### Hybrid-FLiT for sub-millisecond control of sequential neuronal activation

To estimate the temporal resolution for sequential activation of multiple cells or group of cells one must consider the sum of two contributions. First, the illumination dwell time $t_{dw}$ needed to induce a defined light-evoked response in the targeted cell (e.g., a single spike, a train of spikes or a detectable $Ca^{2+}$ response). This, for 2P single-spike generation, is in the range of 1–30 ms and can extend to a few seconds for the generation of multiple spikes[3,10,11,27–32] or for neuronal inhibition[10,33]. Second, the time $t_{sw}$ to switch between distinct light patterns, which corresponds to the refresh time of the LC-SLM (i.e., the time needed by the LC molecules to rotate under a certain voltage and update the phase shift of each pixel). This, with current SLMs, ranges from 3[11] to 16 ms[10,27,30–32].

Here, we demonstrate that FLiT enables sub-millisecond sequential activation independently of the illumination time and LC-SLM switching rate by introducing the principle of *hybrid*-illumination.

Briefly, the idea consists in controlling the relative spiking time of two cells (or group of cells) (e.g., group A and B), with a temporal delay shorter than the necessary illumination dwell time, by using three-phase holograms (Fig. 2a): the first ($\varphi_A$) to generate a light pattern to excite group A, the second ($\varphi_B$) to excite group B and an intermediate hologram ($\varphi_{AB}$) to excite both groups, A and B. By sequentially illuminating the three holograms, using for each hologram an independent illumination time and intensity, it is then possible to sequentially stimulate the two groups of cells with tightly controlled delays, limited by the GM switching time only. Notably, the same principle can be extended to control the relative tuning time between $n$ groups of cells, by using $2n-1$ tiled holograms (Supplementary Fig. 2).

To demonstrate the temporal precision of *hybrid*-illumination, we combined FLiT with a high magnification objective (60×; NA =1) coupled to a fiber amplified laser for excitation (pulse width 150 fs, repetition rate 10 MHz, wavelength $\lambda = 1030$ nm). First, we characterized the effect of the hologram extent on the intensity, ellipticity (i.e., ratio between vertical ($y$) and horizontal ($x$) length) and axial resolution of the generated holographic spot (Supplementary Fig. 3a, b). No significant deterioration of the axial resolution decreasing the tile extent was observed, while we observed a decrease of a factor $\geq 2$ in spot intensity and ellipticity by decreasing the tile vertical extent to $\leq 20$ pixels, corresponding to 30 tiled holograms. For patterns encoding multiple-spots, we found homogenous intensity distribution at the sample plane across 12 tiled holograms (Supplementary Fig. 3c, d). The axial resolution of the spots was preserved both within the field of view (FOV) and between different tiled holograms (full width half-maximum (FWHM) along the axial direction $FWHM_z = 6.5 \pm 0.5$; Supplementary Fig. 3e, f). Spots generated by distal holograms exhibited an axial tilt of $\pm 31°$ which induces negligible deterioration of the resolution ($\frac{FWHM_{tilt} - FWHM_z}{FWHM_z} = 0.12 \pm 0.08$) (Supplementary Fig. 3g, h). Overall, we

could reach homogeneous spots generation across a volume of $120 \times 120 \times 300 \ \mu m^3$ with an axial resolution of $15 \pm 5 \ \mu m$ (Supplementary Fig. 3i–l). To be noted that different optical configurations featuring lower magnification objectives are possible, enabling larger FOVs and number of accessible tiled holograms, as described in the next section.

Next, we measured the minimum switching time between two adjacent holograms with the GM unit (Fig. 2b). For this, we generated 20 equivalent holograms each projecting a single spot on a photodiode placed at a conjugate image plane and we moved the illumination beam across the 20 holograms in discrete steps by driving the galvo with single-step voltage inputs (see details in "Methods"). As a result, we measured a switching time $t_{sw}$ of $90 \pm 10 \ \mu s$ (Fig. 2c, d, $n = 30$ measurements), in agreement with the manufactures specifications. That allowed to switch among multiple patterns at a rate more than one order of magnitude faster than what is achievable by refreshing full frame holograms (Supplementary Fig. 4).

To demonstrate the capabilities of *hybrid*-FLiT for sub-millisecond control of sequential neuronal activation, we combined FLiT with electrophysiological recordings and photoactivated neurons expressing the soma-restricted fast opsin ST-ChroME[10] in acute cortical brain slices while recording cellular activity via whole-cell patch-clamp recordings (see "Methods" and Supplementary Fig. 5). We initially tested different illumination conditions (excitation power and illumination time) which enabled reliable action potential (AP) generation. Consistent with results previously obtained in standard 2P holographic configurations[9,10,27,30,31], APs could be reliably elicited with sub-ms jitter ($0.25 \pm 0.13$ ms; $n = 13$ cells; Supplementary Fig. 6) upon selective targeting of cell somata with $t_{dw} = 4$–5 ms and power $P = 30.5 \pm 13.6$ mW/cell. These values, considering a LC-SLM refresh time of 3–16 ms[10,11], would limit the highest temporal resolution for sequential photostimulation to $t_{dw} + t_{sw} \approx 7 - 20$ ms. To demonstrate that this limit can be reduced by performing *hybrid*-illumination, we photostimulated two ST-ChroME-expressing neurons with different delays while monitoring the evoked activity by double-patch electrophysiological recordings (Fig. 2e). First, we verified that amplitudes and kinetics of induced photocurrents were not affected by switching the illumination between the different holograms (Supplementary Fig. 7a, b). We then assessed the precision in controlling the relative spiking time among the two cells by photoactivating the two patched neurons using *hybrid*-illumination of three holograms imposing tightly controlled delays, $\delta t$, ranging from 0.2 to 3 ms, while measuring the corresponding spiking delay time $\delta t_{AP}^{exp}$ (Fig. 2e, f). As expected, we found that spike delays $\delta t_{AP}^{exp}$ can be controlled with a few hundred μs temporal accuracy ($|\delta t_{AP}^{exp} - \delta t| = 96 \pm 114 \ \mu s$, mean ± SD; $n = 12$ pairs of cells; Fig. 2e, f and Supplementary Fig. 8). It should be noted that the photocurrent magnitude was independent of the vertical position of the tiled hologram (Supplementary Fig. 7c, d). This level of accuracy makes possible to precisely mimic random spiking activity in two distinct neurons. To show this, we photostimulated two neurons with distinct spiking patterns based on physiological activity recorded in vivo (Fig. 2g, inset and "Methods"). Light-driven mimicking was precisely controlled with few hundreds of μs temporal accuracy (Fig. 2g and Supplementary Fig. 9).

Overall, these results demonstrated that *hybrid*-FLiT enables sequential stimulation with a temporal resolution limited to nearly 100 μs, independently of the illumination dwell time and the SLM refresh rate, allowing a precise sub-millisecond tuning of neuronal activity in distinct neurons or groups of neurons.

### Cyclic-FLiT for multitarget excitation

In conventional holographic illumination approaches the simultaneous excitation of $n$ cells is achieved by using a single hologram to illuminate $n$ cells with a constant excitation power $P_{n,std} \approx n \cdot P_{std}$, where $P_{std}$ is the excitation power applied for a time $t_{dw}$ to activate a single

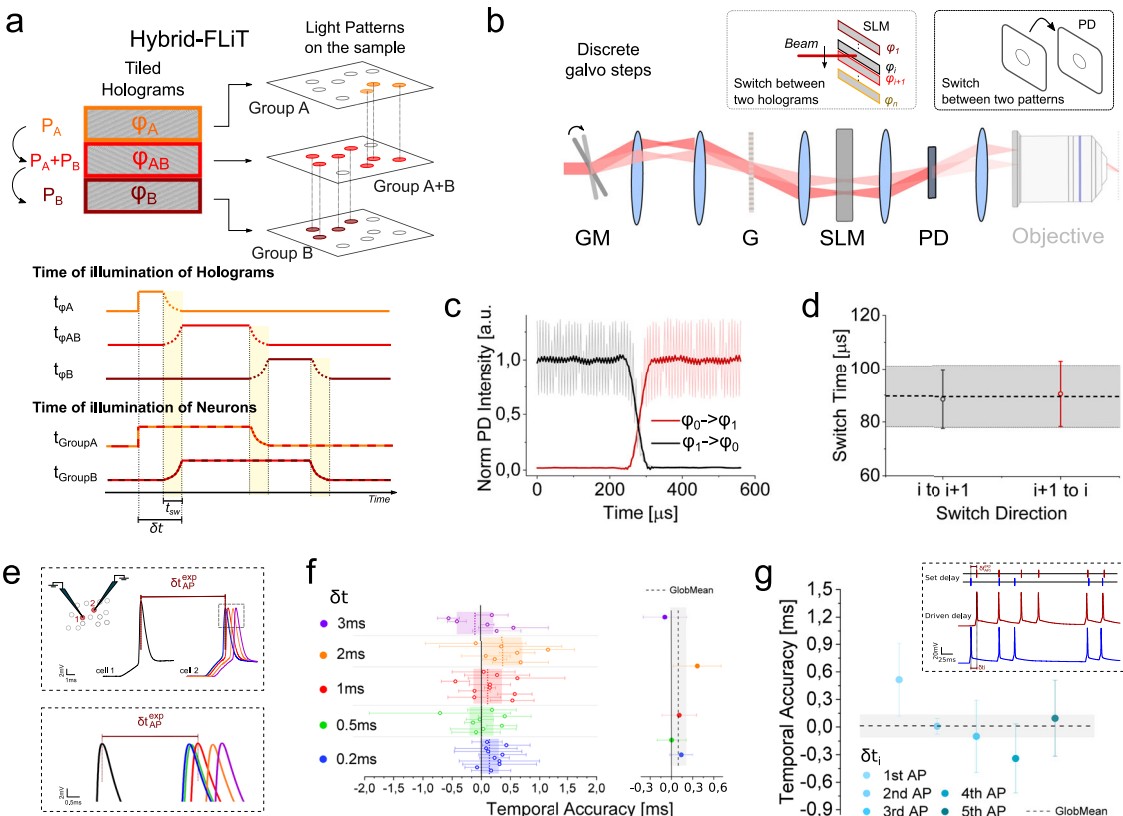

**Fig. 2 | Tuning of neuronal activity in targeted neurons by *hybrid*-FLiT.**
**a** Conceptual scheme of *hybrid*-FLiT. The LC-SLM is tiled in different regions each encoding different phase masks. In the present example, phase mask $\varphi_A$ and $\varphi_B$ encode for group of spots A and B, while phase mask $\varphi_{AB}$ encodes for a comprehensive pattern including group A and group B. By steering the beam vertically across the phase masks with predetermined dwell-times and illumination intensities per each mask, it is possible to set arbitrary delays of activation between groups of spots. In the illustrated example, the illumination dwell time is $t_{\varphi A}$, $t_{\varphi AB}$, $t_{\varphi B}$, and the illumination power is $P_A$, $P_A + P_B$, $P_B$ on the holograms $\varphi_A$, $\varphi_{AB}$, $\varphi_B$, respectively. On comprehensive phase mask $\varphi_{AB}$, the distribution of intensity must be computationally set to maintain an amount of power $P_A$ and $P_B$ on subgroup A and B, respectively. Overall, this scheme yields an activation time $t_{\varphi A} + t_{\varphi AB}$ for group A, $t_{\varphi AB} + t_{\varphi B}$ for group B and a delay of activation between group A and group B $\delta t$ equivalent to $t_{\varphi A}$. $t_{sw}$ represents the switching time of the GM unit. Yellow vertical bands indicate the switching time $t_{sw}$. The scheme displayed is meant to represent *n* groups of spots; their number is here limited to 2 for presentation purposes only. **b** Measurement of the switching time between two adjacent tiled holograms ($\varphi_i$ and $\varphi_{i+1}$) when galvo moves in one discrete step. A photodiode (PD) is placed in an image conjugate plane while driving the galvanometric mirror (GM) with a small-angle single-step voltage input. **c** Representative intensity response of the PD when GM is switched from hologram $\varphi_I$ (encoding for an individual spot in the middle of PD) to hologram $\varphi_O$ (deviating the beam out of the PD) (black line) or, vice versa (red line). SLM was subdivided in 20 holograms. **d** Switch time calculated as the time taken for the signal to rise/fall between 3% and 97% of the maximum intensity, when the spot is encoded in hologram $\varphi_i$ and GM is switched from hologram $\varphi_i$ to $\varphi_{i+1}$ (black symbols) or vice versa (red symbols). Horizontal black line and shaded gray band indicate the global mean and SD switching time,

respectively. **e** Schematics of the experiment for testing the timing of neuronal activity control. Two ST-ChroME expressing patched neurons (cell 1 and cell 2) are photostimulated by using *hybrid*-illumination of three holograms imposing a tightly controlled delays, $\delta t$, ranging from 0.2 to 3 ms, while measuring the corresponding spiking delay time $\delta t_{AP}^{exp}$. Different colors correspond to different delays. Bottom inset: detail of the AP peaks. **f** Left: Temporal accuracy calculated as the difference between imposed $\delta t$ and experimental $\delta t_{AP}^{exp}$ delays, $|\delta t_{AP}^{exp} - \delta t|$. Circle symbols represent different pair of cells activated with $\delta t$ delays (data are shown as mean ± SD). Horizontal dashed line and bands of different colors represent the mean and SD at a specific delay $\delta t$ as indicated on the left, respectively. Error bars are SD on $n = 12$ pairs of cells. Right: Global mean temporal accuracy of all pairs of cells shown on the left of $\delta t$. Different colors correspond to different delays as indicated on the left. The vertical dashed line and the gray band indicate the average and SD temporal accuracy globally calculated for all pairs of cells ($96 \pm 114 \ \mu s$). Data are shown as mean ± SD. Mean photostimulation power is $36.8 \pm 20.9$ mW. Illumination dwell time 4–5 ms. In total, 1030 nm illumination has been used. **g** Temporal accuracy of light-driven random spiking patterns calculated as difference between imposed $\delta t_i$ and experimental $\delta t_{APi}^{exp}$ delays of the *i*th AP pair, $|\delta t_{APi}^{exp} - \delta t_i|$. Circles from light to dark blue indicate temporal accuracy from subsequent pairs of APs (data are shown as mean ± SD). The horizontal dashed line and the gray band indicate the average and SD temporal accuracy globally calculated for all pairs of cells ($n = 6$ pair of cells). Mean photostimulation power is $37.7 \pm 21.3$ mW. Illumination dwell time 2–5 ms. In all the panels, 1030 nm illumination has been used. Inset: Representative light-driven APs from two double-patched ST-ChroME-expressing neurons (bottom) by imposing a random spiking pattern featuring inter-spike-time intervals $\delta t_i$ (top). Source data are provided as a Source Data file.

cell (Fig. 3a). Here, we introduce an alternative illumination protocol based on the principle of *cyclic*-illumination that enables to considerably reduce the total power required for multitarget excitation.

Briefly, if in a time $t_{dw}$, under a constant illumination power $P_{std}$, a given photocurrent (or an AP) is generated in a targeted cell, it can be demonstrated that a similar photocurrent (or an AP) can be produced by using a cycle of $N_{cyc}$ illumination pulses of power $P_{cyc}$ and duration $t_{cyc}$, separated by a time interval $T_{cyc}$ (Supplementary Note 1). The advantage of this configuration is that during the off-time of each cycle

(i.e., $T_{cyc} - t_{cyc}$) the same beam of intensity $P_{cyc}$ can be sequentially redirected to $H = 1/D$ holograms, where $D = t_{cyc}/T_{cyc}$ is the duty-cycle of the *cyclic*-illumination. It follows that *cyclic*-illumination enables simultaneously reaching up to $\frac{P_{std}}{P_{cyc}} \cdot H$ times more cells than conventional holography or the same number of cells using $\frac{P_{std}}{P_{cyc}} \cdot H$ less total power.

To demonstrate this prediction under different experimental configurations, we use the setup described in the previous paragraph to excite, under conventional holography and

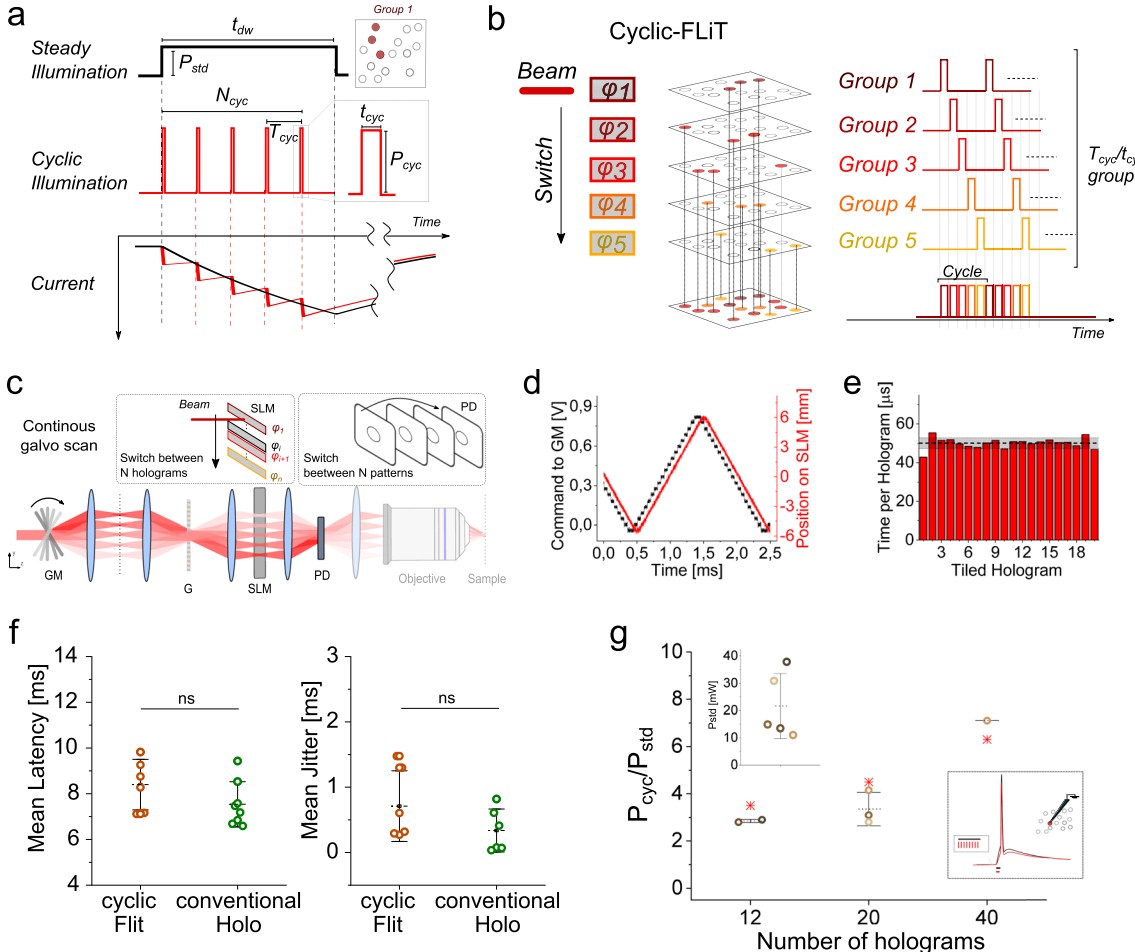

**Fig. 3 | Photoactivation under *cyclic*-FLiT. a** Photostimulation of a group of neurons under steady and *cyclic*-illumination. A soma-targeted light pattern encoded by a single hologram can be used to photoactivate a group of neurons either under steady illumination of power $P_{std}$ and duration $t_{dw}$ (black line, top) or under *cyclic*-illumination of $N_{cyc}$ pulses at power $P_{cyc}$, period $T_{cyc}$ and pulse duration $t_{cyc}$ (red line, middle). Simulated photocurrents generated in a ST-ChroME-expressing neuron are shown under steady (black, bottom) and *cyclic* (red, bottom) illumination when $P_{cyc} = P_{std}\sqrt{T_{cyc}/t_{cyc}}$ ($P_{std} = 0.05$ mW/μm²; $P_{cyc} = 0.05\sqrt{20}$ mW/μm²; $T_{cyc} = 20t_{cyc}$; $t_{cyc} = 50$ μs; 1030 nm). **b** Conceptual scheme of simultaneous photostimulation of multiple groups of neurons under *cyclic*-FLiT. The LC-SLM is tiled in multiple holograms $\varphi_i$ (here from $\varphi_1$ to $\varphi_5$) each encoding for distinct soma-targeted multicell light patterns encoding for different groups of cells (here from Group 1 to Group 5). The illumination beam is switched across the holograms such that the light is sequentially redirected to $H = 1/D$ holograms, and the same cyclic photoactivation process is enabled sequentially on the different light patterns. The scheme displayed is meant to represent $H$ groups of spots; their number is here limited to 5 for presentation purposes only. **c** Measurements of the switching time to sequentially illuminate all holograms at constant rate from $\varphi_1$ to $\varphi_n$ by driving the galvanometric mirror (GM) with a staircase voltage input. **d** GM voltage input (black line) and corresponding position of the incoming beam on the LC-SLM (red line) when GM is driven as depicted in (**c**). **e** Pulse-width dwell time of each hologram $\varphi_i$ of the LC-SLM while GM is driven as depicted in (**c**) and $\varphi_i$ only encodes an individual spot in the middle of the PD. Horizontal black and gray lines indicate the mean and SD dwell time over all holograms, respectively. **f** Mean latency and jitter of the light-evoked APs obtained under *cyclic*-FLiT (brown) and conventional illumination (green) for different number of holograms ($n = 8$ cells; data are shown as mean ± SD). For conventional holography: total illumination time $t_{exp} = t_{dw} = 5$ ms; For *cyclic*-FLiT: total illumination time $t_{exp} = 5$ ms; $t_{dw} = t_{exp}/H$). Data are non-significantly (ns) different between the two illumination protocols ($P$ value = 0.23 and $P$ value = 0.31 for latency and jitter, respectively; Mann–Whitney test, two-tailed). **g** Ratio of the powers needed to trigger a light-evoked AP with 5 ms total time of illumination under steady $P_{std}$ and *cyclic*-illumination $P_{cyc}$ illumination and different number of holograms H. Different colors indicate different cells ($n = 5$ cells; data are shown as mean ± SD). Red asterisks represent the theoretical expected $\sqrt{H}$ ratio value. Top inset: Threshold power to activate the cells under steady illumination with $t_{dw} = 5$ ms. Bottom inset: representative light-evoked APs under steady (black) and *cyclic* (red) illumination of duration 5 ms. Source data are provided as a Source Data file.

*cyclic*-illumination, a ST-ChroME-expressing neuron in organotypic slices (Supplementary Fig. 5). We generated *cyclic*-illumination at variable duty cycles, by dividing the LC-SLM in $H = 1/D$ tiled holograms. We then monitored, using whole-cell patch-clamp recordings, the photo-evoked activity of a targeted patched cell in a conventional scheme, by continuously illuminating the corresponding hologram for a time $t_{dw}$, (i.e., using steady conventional illumination) or in a *cyclic*-FLiT mode, by steering the beam across the $H$ holograms at constant illumination time per hologram, $t_{cyc}$ (i.e., using *cyclic*-illumination of the targeted patched cell with $N_{cyc}$ illumination pulses of duration $t_{cyc}$ every $T_{cyc}$). In this case, the galvo was driven with a staircase function which was found to enable

homogeneous illumination pulses $t_{cyc}$, of $50 \pm 10$ μs (Fig. 3c–e, see "Methods" for details).

We first found the excitation power $P_{std}$, necessary to generate reliable APs under steady illumination for a given illumination time $t_{dw} = 5$ ms ($P_{std} = 20.4 \pm 9.4$ mW; $7.54 \pm 0.9$ ms AP latency; $0.34 \pm 0.33$ ms AP jitter, $n = 8$ cells; Fig. 3f, green dots). We then considered a first configuration for *cyclic*-illumination that could trigger an AP within the same total experimental time, $t_{exp}$ (i.e., $t_{exp} = N_{cyc} \cdot H \cdot t_{cyc} = 5$ ms and cell illumination time $t_{dw}^{cyc} = N_{cyc} \cdot t_{cyc}$). In agreement with the theoretical prediction (Supplementary Notes 1 and 3) we found that this condition is achieved for $P_{cyc} \cong \sqrt{H} \cdot P_{std}$ (Fig. 3g). Importantly, similar ranges of AP latency and jittering where

maintained with this illumination protocol (Fig. 3f, green dots; *cyclic*- and steady-illumination data non-significantly different, $P = 0.23$ and $P = 0.31$, respectively). Critically, redirecting the same beam of intensity $P_{cyc} \cong \sqrt{H} \cdot P_{std}$ to sequentially illuminate $H$ holograms (each generating $m$ spots) will enable reaching a total of $n = m \cdot H$ cells. This compared to the case of conventional holography (1 single hologram for $m$ cells) would represent an overall gain of achievable targets equal to $\sqrt{H}$ times. Alternatively, using a beam of $P_{cyc} \cong \sqrt{H} \cdot P_{std}$ to illuminate $H$ holograms encoding for a total of $m$ targets will enable reaching the same number of targets using $\sqrt{H}$ less total power.

This gain can be further increased by using a second configuration of *cyclic*-FLiT, where conventional holography and *cyclic*-FLiT have the same excitation power $P_{std} = P_{cyc} = P'$ and same cell illumination time $t_{dw}^{std} = t_{dw}^{cyc} = t_{dw}$ (Supplementary Note 3 and Supplementary Fig. 10). Indeed, the photocurrents evoked in each cell sum up at each cycle and, for opsin's turn-off kinetics longer than the period of the cycles ($\tau_{off} > T_{cyc} = H \cdot \tau_{cyc}$), the maximum photocurrent progressively tends to what reached under conventional steady illumination. Thus, *cyclic*-FLiT can in principle reach $H$ times more cells or group of cells, or $H$ times less power for the same number of cells, than conventional holography. Clearly, because with this modality of *cyclic*-illumination the current builds up in a longer time and decays among subsequent pulses, the effective current rise time is slowed down, with consequent lengthening of latency (and jittering). This effect is the more pronounced the higher is the number of used holograms and for $H > 20$, gave rise to a loss of spike generation. Anyhow, incrementing $H$ up to 50 holograms, and thus allowing 50 times more cells, was possible if augmenting the excitation power $P_{cyc}$ by only a factor of $2.48 \pm 0.85$ (Supplementary Fig. 10).

Next, we used *cyclic*-illumination for multitarget excitation on a large FOV. To this end, we built up a FLiT system enabling 2D and 3D multitarget excitation and $Ca^{2+}$ imaging throughout a large FOV (×20 objective NA = 1, mean axial resolution 17–26 μm across $0.35 \times 0.35 \times 1$ mm$^3$; Supplementary Figs. 11 and 12) using a 10 W fiber laser (pulse duration 300 fs, repetition rate 500 kHz, wavelength $\lambda = 1030$ nm). For multiplane $Ca^{2+}$ imaging, the imaging laser (pulse width 100 fs, repetition rate 80 MHz, wavelength $\lambda = 920$ nm) was multiplexed in multi, axially distinct, holographic foci using a second LC-SLM in the imaging path (Fig. 4a and "Methods").

We compared conventional holography with *cyclic*-FLiT illumination, performed by using, as in Fig. 3, the same total experimental time, $t_{exp}^{std} = t_{exp}^{cyc} = t_{exp} = 10$ ms (see also configuration 2 in Supplementary Note 3) to excite $n$ targets so to verify the expected ratio of $\sqrt{H}$ between the total power to excite $n$ cells in conventional holography and in *cyclic*-illumination. For this, we first mapped the locations of $n$ cells co-expressing GCaMP7s and ST-ChroME in organotypic slices (Supplementary Fig. 5). We then calculated $H$ holograms each generating $m$ targets, such that $n = H \cdot m$, and recorded the photo-evoked calcium transients during *cyclic*-illumination of the $H$ holograms. In order to minimize the power losses at the objective back aperture, the LC-SLM was subdivided in $H = 45$ tiled holograms of which we only used the 4, 9, 16, or 23 central ones (see also "Methods"). Importantly, because the tiled holograms are here only used to multiplex a temporally focused Gaussian spot but not to control its shape, the reduced number of pixels per hologram was still enough to generate up to 100 spots without deteriorating the spot quality or axial resolution (Supplementary Figs. 11a and 13). The power per hologram was $P_{n,cyc}$ and each cell was cyclically illuminated with a power $P_{cyc} = \frac{P_{n,cyc}}{m}$ (Fig. 4b). We repeated the same experiment in conventional illumination by continuously illuminating a single hologram encoding for all the $n$ targets ($H = 1$) and increasing the power by a factor of $\sqrt{H}$. The total illumination power was $P_{n,std} = P_{n,cyc}\sqrt{H}$ and each cell was constantly illuminated with $P_{std} = \frac{P_{cyc}}{\sqrt{H}}$. Consistently with the theoretical

expectations, we verified that independently on the numbers of holograms ($H = 4, 9, 16, 23$), these power conditions elicited equivalent ranges of calcium transients (Fig. 4b–d and Supplementary Fig. 14a, b) and similar fractions of responding cells (Supplementary Fig. 14c) under *cyclic*- and conventional illumination (23 cells per FOV; 4 FOVs).

In particular, we found that for $H = 23$ (the maximum number tested in these experimental conditions), *cyclic*-FLiT could reliably activate, either in 2D or 3D volumes, >75% of the targeted cells using a power corresponding to 3 mW/cell (d$F/F = 0.84 \pm 0.96$) or >90% (d$F/F = 2.58 \pm 2.44$) of the targeted cells using 7 mW/cell (Fig. 4e, f, brown bars). At these powers, conventional holography could only generate weak $Ca^{2+}$ transients on a low fraction of cells even at the highest power (activated cells: 35%; d$F/F = 0.25 \pm 0.47$ at 7 mW/cell) (Fig. 4e, f, blue bars). Similar calcium responses and numbers of responding cells were retrieved when incrementing the steady power by $\sqrt{23}$-fold (Fig. 4e, f, green bars). Figure 4g shows an example where we simultaneously targeted 69 cells in a 3D volume ($350 \times 350 \times 60$ μm$^3$) with 215 mW total power with *cyclic*-illumination and $H = 23$ holograms (each illuminating $m = 3$ cells) (Fig. 4g, brown lines) or conventional holographic illumination and $H = 1$ hologram to target 69 cells with 215 mW (Fig. 4g, blue lines) or 1030 mW ($= 215\sqrt{23}$ mW; Fig. 4g, green lines) total power, respectively. Importantly, the use of $\sqrt{H}$ times higher power per cell in *cyclic*-illumination, does not induce a deterioration of the axial resolution with respect to conventional illumination as the saturation power in *cyclic*-illumination scales by the same factor $\sqrt{H}$ (Supplementary Fig. 15 and Supplementary Note 4). Similar in vivo experiments in mouse visual cortex (Supplementary Fig. 16a–c), and zebrafish larva hindbrain (Supplementary Fig. 16d–f), could confirm what demonstrated in vitro that *cyclic*-illumination achieves multitarget stimulation with $\sqrt{H}$ times lower power.

Finally, we evaluated if the need of increasing by $\sqrt{H}$ the local excitation power per cell in *cyclic*-illumination could increase the risk for nonlinear photodamage effects and therefore ultimately limit the number of achievable holograms. To define this limit, we measured the photodamage threshold under our experimental conditions for both conventional and *cyclic*-illumination. We considered the baseline GCaMP fluorescence as an indicator of cell health and monitored the changes in the baseline by prolonged stimulations at different power (Supplementary Fig. 17). In agreement with previous findings[10], we found constant fluorescent baseline for stimulation powers ≤ 170 mW/cell (-0.6 mW/μm$^2$; -0.3 μJ/cell per pulse; -1.3 nJ/μm$^2$ per pulse) for both FLiT and holographic illumination ($n = 115$ cells), ensuring the possibility to safely use up to hundreds of holograms, $H$, before exceeding the nonlinear photodamage threshold (see Supplementary Notes 2).

These findings demonstrate that *cyclic*-FLiT illumination for multitarget excitation needs $\sqrt{H}$ to $H$ less power than using conventional holographic illumination, which enables to reach $\sqrt{H}$ to $H$ more targets than conventional approaches for multitarget optogenetics. The reduction of the global power also enables to minimize thermal damages as it will be further detailed in the following paragraph.

## Cyclic-FLiT illumination enables to minimize thermal effects

*Cyclic*-illumination also reduces the photoinduced temperature rise. To evaluate this property, we used a previously validated heat diffusion model[25,26] to predict the temperature changes generated under different conditions in multicell photoactivation experiments.

We first compared the temperature rise on a single point caused when a spot of 10 μm was steadily illuminated for a time $t_{dw} = 10$ ms and a power $P_{std} = 10$ mW or cyclically illuminated with $N_{cyc}$ short pulses of power $P_{cyc} = \sqrt{H} \cdot P_{std}$, generated by using $H = 20$ holograms ($t_{cyc} = 50$ μs per pulse; $N_{cyc} \approx \frac{t_{dw}}{H \cdot t_{cyc}} = \frac{200}{H} = 10$ pulses). Steady illumination generated a continuously increasing temperature rise which approached its nearly steady-state value at $t_{dw} \approx 10$ ms (Fig. 5a, black line). *Cyclic*-illumination generated a train of fast rising temperature peaks, with each peak corresponding to each cyclic pulse (Fig. 5a, blue

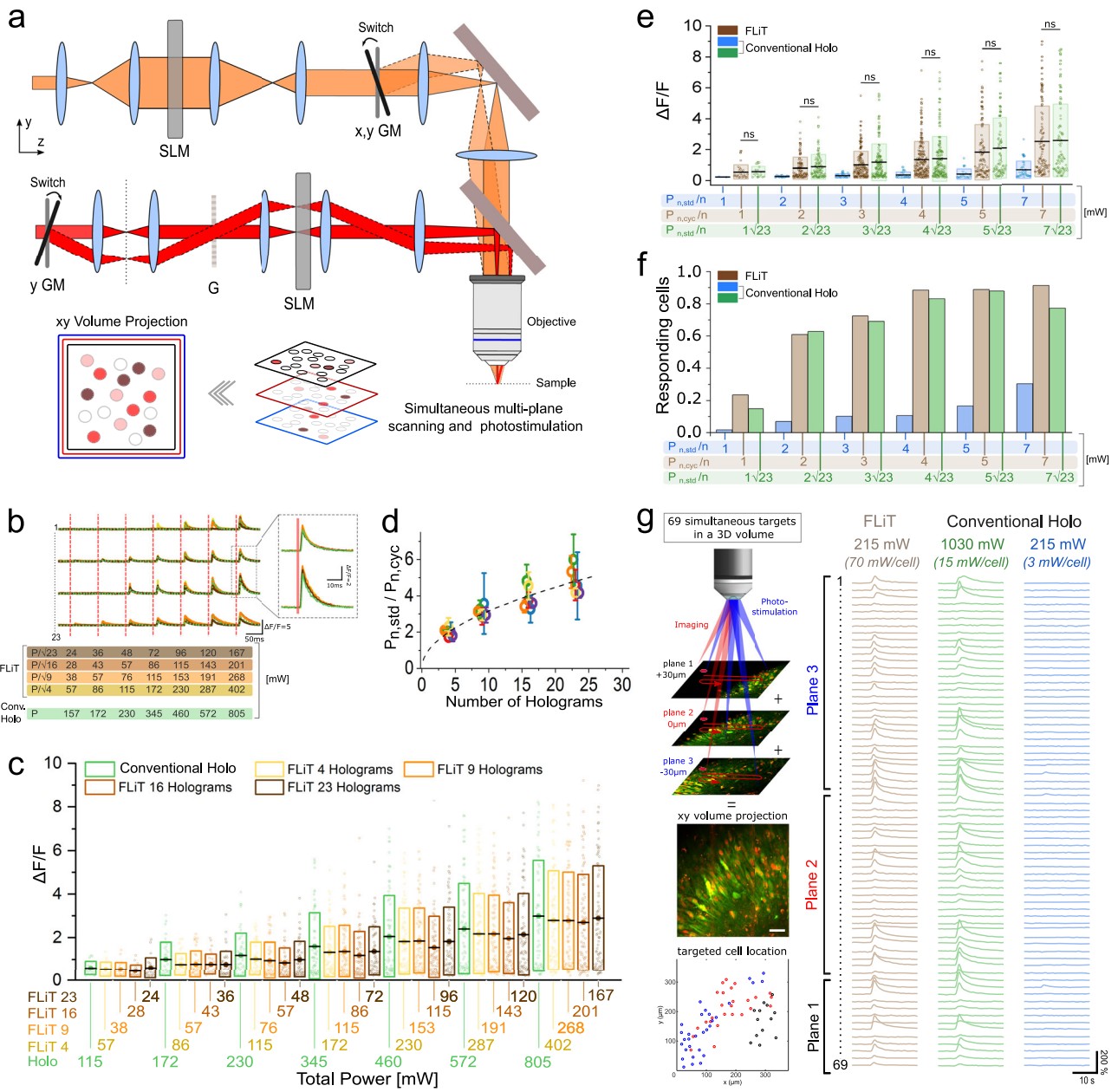

line), which adds to a "background" component (Fig. 5a, red dotted line) related to the thermal accumulation between sequential pulses. Since the rapid fluctuations of *cyclic*-illumination are faster than the characteristic time necessary to produce any thermal effects in the sample[34,35], for the estimation of the total temperature rise in the case of *cyclic*-illumination we only consider this residual heating background. Under these conditions, we found that the more efficient thermal dissipation of *cyclic*-illumination generates a minimal temperature rise compared to steady illumination (Fig. 5a and Supplementary Fig. 18).

We then extended this comparison to the case of multitarget excitation across a volume of $0.35 \times 0.35 \times 0.1 \, \text{mm}^3$ centered around a depth of 150 μm (i.e., corresponding to a photoactivation volume of the mice cortex L2/3) (Fig. 5b). In this case, to compensate for the scattering and keep a uniform excitation per cell of 10 mW, we rescaled, in the modeled light distribution, the power sent per each target ($P_{std}$ or $P_{cyc}$) by the scattering factor: $\chi = e^{z/\ell_s}$, where $\ell_s$ is the scattering length ($\ell_s = 166$ μm for a wavelength of 1030 nm; see "Methods") and $z$ the target depth. At first, we simulated the temperature rise at the center of the volume for 160 spots. In these

conditions, in addition to the temperature rise due to direct illumination, a slow contribution arises from the other targets bringing the total temperature rise to ~5 K after 10 ms of illumination (Fig. 5c, black line). In the case of *cyclic*-illumination ($H = 20$ holograms, generating 8 spots each), the use of $\sqrt{H}$ lower power significantly reduces the contribution from neighboring targets and gives rise to a temperature increase of about 1.2 K, i.e. nearly five times lower than the one generated using conventional holography (Fig. 5c, blue line). Same results hold for different spot density and illumination times (Fig. 5d–f).

All in all, these simulations indicate that the use of *cyclic*-illumination reduces the temperature rise both for single-cell excitation because of the use of sequential brief pulses which enables more efficient thermal dissipation than a long steady pulse, and, for multitarget configuration because of the use of $\sqrt{H}$ less total power.

## Discussion
### Achievements
We have demonstrated an optical configuration for ultrafast sequential light targeting (FLiT), where a galvanometric mirror enables fast

**Fig. 4 | Multicell all-optical *cyclic*-FLiT. a** Scheme of the optical setup including: a multiplane imaging system (orange path) and a *cyclic*-FLiT photoactivation system (red path). Multiplane imaging relies on a SLM-based modulation of the incoming laser which splits the laser in multifoci beams simultaneously scanning axially-shifted planes (red, blue, and black planes). Multiple cells can be independently photoactivated in multiple-planes (red, light-red, and dark-red circles, limited to three planes for representation purposes). GM: Galvo-mirrors, G: diffraction grating. **b** Representative calcium traces from 4 neurons of a group of 23 co-expressing GCaMP7s and ST-ChroME cells simultaneously photoactivated by varying the illumination mode (*cyclic*-illumination with $H = 4$ (yellow line), $H = 9$ (orange line), $H = 16$ (light brown line), $H = 23$ (dark brown line) holograms or conventional steady holography ($H = 1$, green dashed line) and the illumination powers. Vertical red lines indicate the onset time of each photostimulation episode. The power corresponding to each photostimulation episode is indicated at the bottom with rows of different colors, each indicating the power used in the different illumination modes. Power was adapted such that the total power of *cyclic*-FLiT was reduced by a factor equal to $\sqrt{H}$ compared to conventional holographic illumination. The inset represents a zoom of a part of the calcium traces. For conventional holography: total illumination time, for $n = 23$ cells, $t_{exp} = t_{dw} = 10$ ms; For *cyclic*-FLiT: total illumination time, for $n = m \cdot H = 23$ cells, $t_{exp} = 10$ ms; $t_{dw} = t_{exp}/H$. **c** dF/F upon multicell all-optical photoactivation based on the illumination protocol depicted in (**b**). Different colors indicate different illumination modes as indicated in (**b**). Illumination powers corresponding to different illumination modes are indicated along the *x* bottom axis. Box bounds and box center indicate standard deviation and mean of the dF/F of all cells photoactivated in each illumination mode, respectively (4 FOV, 365 × 365 μm², 23 cells simultaneously photoactivated per FOV). Circles indicate the dF/F of each cell. **d** Ratio of the total illumination power needed in conventional steady illumination and *cyclic*-FLiT ($\frac{P_{n,std}}{P_{n,cyc}}$) to induce the same range of dF/F in the photoactivated cells for different numbers of holograms. Circles and

bars indicate the mean and the standard deviation of the responding photo-activated cells binned on different ranges of dF/F (0.3<dF/F<0.5 blue, 0.5<dF/F<1 green, 1<dF/F<2 red, 2<dF/F<3 orange, 3<dF/F<5 purple, dF/F>5 yellow). Black dashed line indicates the theoretical $\sqrt{H}$ factor. $n = 23$ cells per FOV; 4 FOVs. **e, f** dF/F (**e**) and fraction of responding cells (**f**) upon multicell all-optical photoactivation under *cyclic*-FLiT with $H = 23$ holograms (dark brown) and steady conventional illumination (blue) by keeping the same power per cell or by increasing the power of conventional illumination by $\sqrt{H} = \sqrt{23}$ (green). Box bounds and box center indicate standard deviation and mean of dF/F of all cells photoactivated in each illumination mode, respectively (Kruskal–Wallis test followed by Dunn's multiple comparison; ns: $P > 0.05$; 4 2D-FOV, 350 × 350 μm², 92 cells; 3 3D-FOV, 350 × 350 × 60 μm³, 115 cells). Circles indicate dF/F of each cell. **g** Left: Schematics of multifoci 2P scanning image of a set of 69 cells simultaneously photoactivated and recorded in a 3D volume. Cells are in three distinct planes 30 μm axially apart. Cells belonging to different planes are simultaneously monitored in a 2D *XY* volume projected image. Red and green corresponds to ST-ChroME and GCaMP7s labeling, respectively. The locations of the 69 photoactivated cells distributed across the three planes is indicated with different colors in the 2D map at the bottom (black, red and blue circles correspond to cells located in $z = 30$, $0$, $-30$ μm plane, respectively). Scale bar: 50 μm. Right: Calcium transients associated to the 69 cells located in a 3D volume as depicted on the left, simultaneously photoactivated and recorded under *cyclic*-illumination with $H = 23$ holograms or conventional steady illumination (215 mW total power under *cyclic*-FLiT (brown line) and conventional illumination (blue lines) or $1030 \approx 215\sqrt{23}$ mW total power under conventional illumination (green lines)). The corresponding powers sent to each cells are also reported. For conventional holography: total illumination time, for $n = 69$ cells, $t_{exp} = t_{dw} = 10$ ms; For *cyclic*-FLiT: total illumination time, for $n = (m \cdot H) = 69$ cells (with $H = 23$ and $m = 3$ cells per hologram) $t_{exp} = 10$ ms; $t_{dw} = t_{exp}/H$. Source data are provided as a Source Data file.

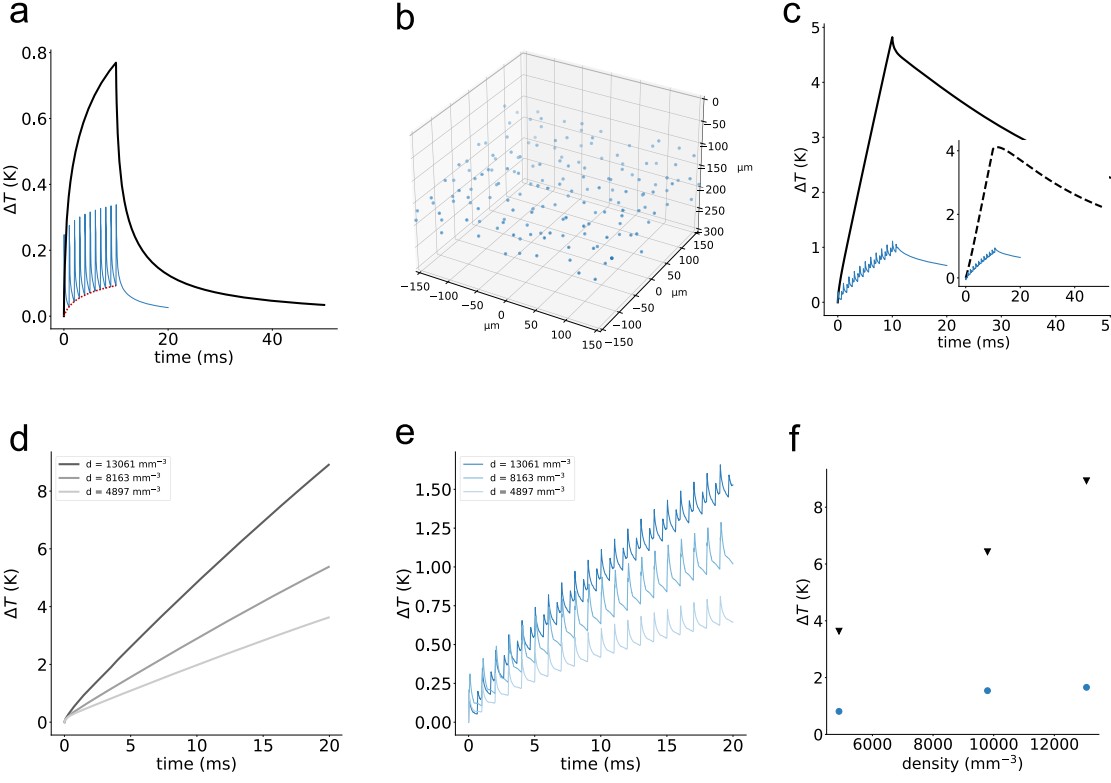

**Fig. 5 | Simulated temperature rise under cyclic and conventional holographic illumination. a** Temperature rise induced on an illuminated spot under conventional (black) and *cyclic*-illumination (blue), using a power per cell, $P_{std}$, of 10 mW and $P_{cyc} = \sqrt{H} \cdot P_{std} = \sqrt{H} \cdot 10$ mW, in conventional and *cyclic*-illumination, respectively; illumination $t_{dw} = 10$ ms; $H = 20$; 50 μs illumination pulses; 1 ms per each cycle **b** Volumetric distribution of 160 spots ($H = 20$ holograms, $m = 8$ spot per hologram) uniformly distributed in a $350 \times 350 \times 100$ μm³ volume. **c** Temperature rise induced on a central spot when 160 spots are illuminated as depicted in (**b**) under conventional (black) and *cyclic*-illumination (blue). Inset: Temperature rise induced on the central spot by the 159 neighboring spots. **d, e** Temperature rise for different spots density $d$ distributed in the considered volume under conventional (**d**) and *cyclic*-illumination (**e**). **f** Maximum temperature rise for *cyclic*- and conventional-illumination as simulated in (**d, e**). $t_{dw} = 20$ ms. Power per cell in conventional illumination $P(z) = 10 \cdot e^{z/\ell_s}$ mW; Power per cell in *cyclic*-illumination $P(z) = 10 \cdot e^{z/\ell_s} \cdot \sqrt{H}$; $H = 20$ in (**b–d**).

switching through multiple vertically tiled holograms. We have used this system to demonstrate two new illumination paradigms for multitarget optogenetics, *hybrid*- and *cyclic*-illumination, which enable sub-millisecond control of sequential neuronal activation and high-throughput simultaneous multicell illumination, respectively.

Optical control of multiple neurons generally requires holographic light multiplexing through the use of LC-SLMs either coupled with spiral scanning or with scan-less patterned illumination[3,21]. In these configurations, the temporal resolution for sequential light patterning is limited by the LC-SLM refresh rate (300–60 Hz) at about 3–20 ms. Faster switching times (2 ms) can be achieved using the LC-SLM in overdrive mode and at 85% of its diffraction efficiency[11].

Optical generation of a single AP using powers close to the saturation power have been reached with illumination times below 1 ms[11,28–30]. However, optimal axial resolution is not achieved when working at saturation, this requires working at lower power[31] which typically lengthens the illumination time to 5–30 ms[3], for single-spike generation, or a few seconds for the generation of multiple spikes[11] or for neuronal inhibition[10,33]. These dwell times impose an intrinsic additional temporal delay to control the activity of sequential neuronal patterns.

We have shown that with *hybrid*-FLiT, the sequential illumination of three holograms with a temporally focused beam enables fine control of the relative spiking time between two cells (or groups of cells) with a temporal delay limited by the switching time of the GM unit (in our case about 100 µs), independently of the cell illumination dwell time. Clearly, to fully exploit its high-temporal precision, *hybrid*-illumination needs to be combined with illumination conditions (parallel illumination, fast-opsins, short illumination time) that assure sub-millisecond jittering[10,30,31].

We also have demonstrated FLiT combined with *cyclic*-illumination, termed *cyclic*-FLiT, where $H$ holograms are cyclically illuminated by steering a temporally focused line between holograms at a constant time per hologram. In *cyclic*-FLiT, the cell or the group of $m$ cells encoded by each hologram can be cyclically excited with a series of short pulses (in our case of 50 µs). We have shown, theoretically and experimentally, that this illumination protocol can simultaneously activate a single or $m$ neurons with the same photocurrent and efficacy of conventional steady illumination, provided that each neuron or group of $m$ neurons is cyclically excited with a power $P_{cyc} = \sqrt{H} \cdot m \cdot P_{std}$ with $P_{std}$ being the power necessary to excite each neuron under steady illumination. We have demonstrated both in vitro and in vivo, that under these conditions, it is possible to use the off-time of each cycle to redirect the same light to $H$ distinct holograms thus exciting a total of $H \cdot m$ cells using $\approx \sqrt{H}$ (here demonstrated up to ~5) times less power than in conventional holography.

These findings have two main implications as discussed in the following paragraph: given a defined laser power, the achievable number of targets can be increased by $\approx \sqrt{H}$ times and, given a defined cell distribution, the sample heating can be reduced by $\approx \sqrt{H}$.

Two-photon optogenetics is typically performed using excitation wavelengths in the range of 920–1060 nm. High repetition lasers (80 MHz repetition rate; ≤150 fs pulse width; 920–950 nm), compatible with both photostimulation and fast functional imaging, provide up to 1–4 W of exit power (typically corresponding to about 200–800 mW at the objective exit). Reliable neuronal activation at these wavelengths has been reached using 30–50 mW/cell[33], which limits multitarget stimulation to a few cells. Lower repetition-rate fiber lasers (100 kHz to 10 MHz, typically at 1030–1064 nm[11,20,29,31] or 920 nm[36]) have higher peak powers which enables a reduction in the power required for photostimulation to about 1–20 mW/cell and can deliver up to 60 W and thus potentially activate several hundreds of cells. The true number of targets that can be achieved in this case depends on several parameters such as opsin efficiency, expression level, cell-type and, above all, illumination depth. The latter requires

increasing the power per target by $\sim\chi = e^{z/\ell_s}$ corresponding, in mouse brain, to a factor of ~2 to 20 for depths comprised between 100 and 500 µm. This correspondingly decreases the number of accessible targets. For shallow depths or low scattering samples, high powerful lasers give access to several hundreds of targets, however, under these conditions, the high cell density would lead to major temperature cross-talks possibly reaching levels above the thermal photodamage threshold[25]. Precisely, it has been shown that small temperature variations (>2 K) can affect ion channel kinetics and conductance[37], synaptic transmission[38] and neuronal firing rates[39] across various brain regions. Recent studies have revealed that 10 s visible light application in the murine dorsal striatum at $P = 15$ mW (corresponding to a temperature rise of ~2 K) was sufficient to inhibit neural activity and cause a biased turning behavior[40]. Importantly, there can be a dissociation between physiology and behavior as changes in temperature can induce physiological changes in the absence of detectable changes in behavior[41]. Concerning 2P optogenetics, it has been reported that under typical conditions for in vivo photostimulation (using spiral scanning or whole-cell illumination; 3–40 ms illumination time, 2–50 mW, excitation power, 1030 nm excitation wavelength, 0.5–10 Hz repetition rate), single-cell excitation only induces a temperature rise in the range of few tenths of a Kelvin for a brief period[25]. While in this case the thermal photodamage can be assumed as negligible, important effects can occur in multitarget excitation of dense cell populations. This can generate long-lasting temperature rise of several Ks comparable to what has been reported for 1P excitation and would ultimately also limit the maximum achievable number of targets. Taken together, these factors have so far limited the maximum number of simultaneously photostimulated cells optogenetics to a few tens[11,29].

We have demonstrated that *cyclic*-illumination enables photostimulation of $H \cdot m$ cells using $\approx \sqrt{H}$ (here demonstrated up to ~5) times less total power than in conventional holography, without affecting axial resolution or nonlinear photodamage effect. This gain will therefore allow the number of achievable targets to be increased by the same factor.

This gain can be further increased by using for conventional holography and *cyclic*-FLiT the same excitation power $P_{std} = P_{cyc} = P'$ and same cell illumination dwell time $t_{dw}^{std} = t_{dw}^{cyc} = t_{dw}$. In this case, *cyclic*-FLiT can in principle excite $H \cdot m$ cells using $H$ times less power than conventional holography. Specifically, using the fast opsin ChroME and this last configuration, we could prove a gain of H-20 times. Reaching a larger gain (and therefore a larger number of holograms) will require using opsins with slower off-kinetics (C1V1[42], ChRMine[11], or ReachR[43]) so to compensate for the current losses during the pulse time interval $T_{cyc}$.

We have here compared *cyclic*-FLiT with conventional holography, choosing for conventional holography an illumination time sufficiently long to generate a spike with powers $P_{std}$ lower than saturation so to avoid for conventional holography and *cyclic* important degradation of the axial resolution. For values of $P_{std}$ close or higher than saturation, *cyclic*-FLiT will still enable a gain from $\cong \sqrt{H}$ to $H$ times depending on the configuration used. However, both approaches will have a deteriorated axial resolution and increased risk for nonlinear photodamage effects.

An additional case that can be compared with *cyclic*-FLiT is conventional holography using continuous illumination for a dwell time $t_{dw} = N_{cyc} \cdot T_{cyc}$ and high power, $P_{std}^{us} = kP_{std} \geq \sqrt{H}P_{std}$. In this case, for a given total power budget, the number of achievable cells reachable with conventional holography will be reduced by the same factor $k$. This is opposite to the case of *cyclic*-FLiT where the same dwell time is reached by *cyclic*-illumination with short pulses of duration $\tau_{cyc}$ every $T_{cyc}$ and the time interval $T_{cyc}$ among two sequential pulses is used to visit $H$ more holograms so to reach $H$ times more cells.

As for thermal damages, *cyclic*-illumination enables a considerable reduction in the temperature rise compared to conventional steady multitarget excitation. This is the consequence of two effects: first, at the level of the single cell, the use of sequential brief pulses (50 μs) enables more efficient thermal dissipation than a long steady pulse. Second, the $\sqrt{H}$-fold decrease of the total power used enables to accordingly reduce the global heating in multitarget configuration compared to conventional holography. This will be of particular relevance for multitarget excitation of dense population of cells in small living organisms (as e.g., the zebrafish larvae)[44].

All in all, the main advantage of *cyclic*-illumination in combination with low-power laser or with high-power laser, in experimental designs aiming at reaching multiple sparse targets in depth, will be to increase of $\sqrt{H}$ to $H$ fold the number of targets; for experiments using high-power laser at shallow depth (and therefore not limited in the theoretically achievable cell numbers) it will enable keeping multitarget excitation under safe thermal ranges. Clearly, to push *cyclic* multitarget excitation to the limit of several hundreds of cells will require parallel progress in targeting strategies that optimize expression volume and opsin-indicators co-localization. New transgenic lines and GCaMP-opsin co-expressing virus are gradually becoming available[45] to the community and will certainly enable soon to reach this upper limit.

Previous strategies to increase the number of achievable cells have lowered the laser repetition rate[7,20,27,31]. To be noted that, in this case, increasing by $\sqrt{H}$ the number of reachable targets requires lowering the laser repetition rate of $H$. This will be possible up to a minimum repetition rate below which important loss in average power or pulse broadening will counterbalance the expected benefits. For example, reaching the same power gain of $\sim\sqrt{23}$, here demonstrated with *cyclic*-illumination with a 500 kHz laser, would require keeping the same average power and lowering the repetition rate down to 20 kHz, which is today technically out of reach. Despite this limit, once reached the minimum repetition rate enabled by a specific laser, *cyclic*-illumination used with the same repetition rate will still enable to further lower the total power by $\sqrt{H}$ and consequently reduce thermal damages.

## Comparison with other approaches

Few configurations have previously proposed the combination of a galvanometric system with spatial light modulators or the use of vertically tiled holograms[11,23,46,47]. However, they were designed for different applications and/or their optical designs are not compatible with *cyclic*- or *hybrid*-illumination.

Specifically, in ref. [23], we proposed the use of two LC-SLMs to produce multiple shaped temporally focused patterns, where both SLMs were addressed with vertically aligned rectangular holograms. The system did not include a galvanometric system, therefore the switch rate among successive holograms was limited by the LC-SLM and the cell dwell time, and the maximum number of achievable targets was limited to $\frac{P_{max}}{P_{std}}$ (with $P_{max}$ maximum available power), as in conventional holography.

Yang et al.[47] proposed to couple one LC-SLM with a system of galvanometric mirrors to increase the accessible FOV. Basically, two galvanometric mirrors conjugated to the LC-SLM allowed to laterally shift groups of diffraction-limited spots beyond the FOV typically reached in holographic illumination. In this scheme, multiple targeted points spanning over large extensions could then be excited in a so-called "time-division multiplexing" excitation strategy, where multiple subsets of targets are sequentially illuminated. While time-sequentially separating the excitation allows an overall increment of the total averaged 2P signal compared to fully parallel illumination, that comes at the price of a loss of simultaneity of excitation. In addition, this scheme is limited in terms of timing, as the switch of patterns is bounded to the SLM refresh rate, and in terms of spatial flexibility, as

incapable of producing axial confinement for extended patterns. To ease on temporal limitations, Sun et al.[46] proposed similar concepts by using holograms (up to 6) generated side-by-side on the LC-SLM display and one further galvanometric mirror to switch among them. In particular, the first galvo rapidly switches between different tiled holograms on the LC-SLM (each generating multiple dots on a $120 \times 120\,\mu m^2$ FOV), while the second galvo rapidly shifts the patterns generated by each hologram on the sample plane, thus enlarging the achievable excitation region ($0.2 \times 1.3\,mm^2$). While gaining in temporal resolution, that scheme further compromises spatial flexibility as it reduces the number of pixels per holograms (which need to encode in this case both for the shape and the location of the illumination patterns) and critically compromise the generation of extended 3D shapes. Finally, the designs by Sun et al.[46] and Yang et al.[47] allowed generation of static 2P points-like illumination, which are not immediately compatible with 2P optogenetics except if providing a further galvo to quickly scan the focal points over the cellular bodies. Opposite to that, FLiT facilitates 2P photoactivation as it includes shaped temporal focusing patterns, which enables full flexibility for axially confined illumination patterns, and relies on a spatial multiplexing scheme, where the holograms only need to encode for the $x$, $y$, $z$ locations of the spots, largely reducing the amount of pixels needed (we demonstrated that 3D patterns of spots could be produced even by dividing the SLM with more than 40 side-by-side holograms). Moreover, sub-ms temporal tuning of neuronal activity of several groups of neurons is achieved by operating FLiT in *hybrid*-illumination, incompatible with previous designs. Finally, *cyclic*-FLiT illumination overcomes the temporal restrictions of "time-division multiplexing" by enabling an increment of the 2P signal while simultaneously activating multiple cells.

To this date, the fastest scheme for sequential 2P optogenetics have been demonstrated in ref. [11] by using two fast LC-SLMs operating in overdriving mode and combined with a couple of galvos to enable spiral scanning and 2P optogenetics. Holographic patterns were updated every 2 ms on each SLM by halting the holographic reconstruction to the 85% of its total efficiency, thus causing the loss of nearly 30% of the 2P excitation. In these conditions, if using an ultrafast optoelectronic switcher (switching time $t_{sw} \approx 80\,\mu s$) between the SLM, it is possible to alternate between two fixed holograms within 80 μs or achieve nearly kHz hologram updating. This scheme however might present limitations when it comes to achieve sub-ms tuning of neuronal activity of multiple neurons as proposed in *hybrid*-FLiT or the *cyclic*-illumination of multiple groups of neurons as proposed with *cyclic*-FLiT as in this case respectively $2H-1$ or $H$ independent SLMs in cascade should be used.

A configuration using a galvo system to scan an horizontally focused illumination line through vertically tiled hologram has been recently proposed by Cohen et al.[48] for 2D ultrafast scanning of a diffraction-limited spot. In its preliminary demonstration, the system uses vertically 1 pixel size holograms, it is therefore non-suitable for the generation of extended patterns for 2P optogenetics or the *hybrid*- and *cyclic*-illumination schemes proposed in this manuscript.

## Current limitations and technical outlook

In the current implementation of FLiT, we have shown that we can tile the LC-SLM with up to 45 independent tiled holograms, out of which we can use the central 23 without significant deterioration of spot quality, axial resolution, or light efficiency. Including a de-scanning unit, so that each scanned hologram is projected at the center of the objective back aperture independently of its position on the LC-SLM will enable to eliminate the axial tilt and intensity losses for spots generated with distal holograms (out of the 23) and therefore to use all the 45 holograms.

To be noted that this high number of holograms was reachable because, opposite to previous configurations[23,46], tiled holograms are

here only used to multiplex a temporally focused Gaussian spot but not to control its shape, therefore requiring far fewer pixels.

In the present configuration using a 20x objective we have demonstrated an optical axial resolution of 17–26 µm across $0.35 \times 0.35 \times 1\,mm^3$, depending on the SLM and objective back aperture illumination. These values are comparable with previously reported values using complementary configurations as 3DSHOT (~20 µm[10]), holographic spiral scanning (14–22 µm[11,29]), multitarget holography (20–25 µm[20,36]), for in vivo optogenetics at cellular resolution. Importantly, we have demonstrated that the physiological resolution is the same in conventional and *cyclic* illuminating despite the need of using $\sqrt{H}$ higher power per cell as the saturation power under *cyclic*-illumination rescales by the same factor $\sqrt{H}$. This configuration will therefore enable to achieve the same precision achieved in previous demonstration of in vivo 2P optogenetics.

As shown in this manuscript, the implementation of the FLiT configuration into an existing system using 3DSHOT requires only the addition of a galvanometric mirror before the grating for temporal focusing. The implementation of FLiT with holographic spiral scanning is also relatively straightforward as it will only require to add one (or two) galvanometric mirrors to scan in one (or two) directions the multiplexing SLM array synchronously with the movement of the galvo used for spiral scanning of the spots at the sample plane. The combination of FliT with holographic light-shaping systems[6] will require to introduce an additional asymmetric focusing unit (e.g., a cylindrical lens) in order to produce tiled illuminations on the multiplexing LC-SLM). Despite the increased complexity, this configuration will have the advantage of generating spots of variable size and shape.

We have here used a galvanometric mirror as the switching unit. Different types of scan unit, such as polygonal scanners or AODs could be incorporated to further improve the speed of the switch between light patterns. In the current work, we employed pulsed lasers featuring a repetition rate as low as 500 kHz. Given the switch time of the galvanometric mirror (50–100 µs) it was not essential a synchronization between pulse rate and galvo movements. Such a synchronization would become anyhow important if laser at 100 kHz or lower are employed.

In the present experiments, we limited the use of *cyclic*-FLiT to tiled sub-hologram belonging to one single SLM frame. It would be possible to increment the number of tiled holograms in the limit where the SLM refresh rate equals the time needed to scan the H holograms.

## Applications

Sub-millisecond control of sequential neuronal activation demonstrated using *hybrid*-illumination opens the way to the investigation of synaptic integration, connectivity, and neuronal coding with unprecedented temporal precision. The ability to rapidly switch between multiple photostimulation patterns with sub-millisecond resolution will enable precise investigation of spatial- and time-dependent synaptic summation and integration of multiple and complex synaptic inputs[49]. Being able to stimulate multiple specific subsets of neurons, with single-cell precision, either simultaneously or with sub-ms custom temporal delays will be essential to precisely probe mechanisms such as spike-time-dependent plasticity (STDP), where the temporal interval between pre-and-postsynaptic spikes are necessary to strengthen or depress synaptic connections[50–53].

Previous studies on mammalian neocortex have shown that optogenetic manipulation of small (≤30 cells) groups of neurons appears sufficient to impact behavioral responses[11,12] and most importantly that this can depend on the relative degree of synchronicity among the optically evoked spikes[20,54]. FLiT combined with *hybrid*-illumination has the potential to refine this type of studies by mimicking with unprecedent fine temporal precision a variety of physiological firing patterns and to manipulate them with different flavors, synchronizing or de-synchronizing them at will, while observing the effect of this time-controlled manipulation at different levels, from the local response of a neuronal circuit to behavioral responses and sensory perception, in both healthy and pathological brains. To really take advantage of the superior temporal resolution here demonstrated for sequential photostimulation, it will be necessary to reach the same temporal precision in multicell spike recording, for example with the development of efficient genetically voltage indicators for 2P multitarget voltage imaging.

While we have here shown that *cyclic*-illumination enables a factor of 4–5 increase of the achievable number of targets using a high-power low-repetition 1030 nm laser, the same principle can be applied in combination with different types of lasers. Of particular interest is the use of *cyclic*-illumination with laser system having limited exit power such as mode locked Ti:Sapphire lasers routinely used in most microscopy lab. The use of these lasers offers multiple advantages: the 900–950 nm wavelength range has the advantage of enabling photostimulation of blue-shifted opsins (PsChR2[29], TsChR2[55], CoChR[55]) at their optimal photostimulation peak and for all-optical experiments to combine multitarget photostimulation of these opsins with imaging of red $Ca^{2+}$ indicators which drastically reduces optical crosstalk from the imaging laser[15,33]. Alternatively, they can be used as a single source in all-optical experiments where the photostimulation laser is also used for GCaMP imaging[14]. On the other side, these lasers can typically deliver up to 4 W in the 900–950 nm range and are therefore not sufficient for in depth multitarget cell stimulation. Combining these excitation sources with *cyclic*-illumination will enable a virtual increase of a factor of 4–5 of the power available for photostimulation.

On the other hand, *cyclic*-FLiT for two-photon optogenetics combined with high-power (>50 W) low-repetition lasers opens up the possibility to control large neuronal populations and so to extend all-optical circuits manipulation to the investigation of large neuronal circuit or to more complex mammalian models.

In conclusion, FLiT illumination is a powerful tool for the investigation of neuronal circuits with a sub-millisecond control, at single or large neuronal population scales. Combining all the aspects of FLiT presented here, together with the latest engineered fast activity sensors, will allow an all-optical interrogation and manipulation of brain activity to decipher how specific spatiotemporal patterns produced on user-defined neuronal ensembles influence specific behaviors, cognitive tasks or defined pathological conditions.

## Methods
### Optical setup
The optical system characterized in Supplementary Fig. 3 and used for the results reported in Figs. 2–3 was built around a commercial upright microscope (Olympus BX51WI) placed on a *XY* stage for sample displacement (Luigs & Neumann, V380FM). A femtosecond pulsed beam delivered by a diode-pumped, fiber amplifier system (Amplitude Systems, Goji HP; pulse width 150 fs, tunable repetition rate 10–40 MHz, maximum pulse energy 0.5 µJ, maximum average power 5 W, wavelength $\lambda = 1030\,nm$) operated at 10 MHz, was sent first through a $\lambda/2$ wave retarder (Thorlabs, 690–1200 nm, AQWP05M-980) in combination with a polarizer cube (CVI Melles Griot) for a manual control of the laser power. The beam was then demagnified with a telescope (f1 = 100 mm; AC508-100-B, Thorlabs; f2 = 50 mm, AC508-50-B, Thorlabs) and sent through an acousto-optic modulator (AOM) (AA Opto-Electronic, France) to drive fast and precise light power control. The first diffracted order was projected on a pair of XY GMs (3 mm aperture, 6215H series; Cambridge Technology) with a de-magnifying telescope, $T_{beam}$ (M = 0.4 magnification). Only the Y GM was used and driven by a servo driver (Cambridge Technology, MicroMax series 671).

The GM plan was conjugated to a reflective dispersion grating of 800 l/mm by means of a telescope, $T_{GM-G}$, (f = 250 mm; AC508-250-B, Thorlabs; f = 500 mm, AC508-500-B, Thorlabs). A lens ($f_L = 500\,mm$,

Thorlabs, AC508-500-B) transmitted the resulting spatially chirped beam on the sensitive area of a reconfigurable liquid-crystal-on-silicon LC-SLM (LCOS-SLM X10468-07, Hamamatsu Photonics, resolution $800 \times 600$ pixels, 20-µm pixel size), located in the Fourier plane of the diffraction grating. The LC-SLM was finally conjugated to the back focal plane of the microscope objective (Olympus LUMPlanFL 60XW NA 0.9) via a telescope, $T_{SLM\text{-}obj}$, ($f = 1000$ mm; AC508-1000-B, Thorlabs; $f = 500$ mm, AC508-500-B, Thorlabs).

The size of the beam, $D_{beam}$, at the Y GM depends on the desired spot size at the focal plane, $D_{Spot}$ and the objective used: $D_{Spot} = \frac{M_{T(GM-G)}}{M_{T(SLM-obj)}} \cdot \frac{f_{Obj}}{f_{SLM}} \cdot D_{beam}$, where $M_{T(GM-G)}$ and $M_{T(SLM-obj)}$ are the magnifications of the telescopes between GM and Grating and between SLM and objective back focal plane respectively, $f_{SLM}$ is the focal length of the focusing lens between grating and SLM and $f_{Obj}$ the focal length of the objective lens. For the two objectives used in this manuscript, we measured spot sizes of 8 and 13 µm (FWHM of the 2P fluorescence profile) corresponding to a $D_{beam}$ always smaller than the 3 mm size of the GM.

The system used for the all-optical experiments reported in Fig. 4 had a similar architecture adapted to be coupled to a Zeiss microscope (Axio Examiner.Z1), a Zeiss W Plan-Apochromat 20× NA 1.0 objective and a $1920 \times 1152$ pixels LC-SLM (HSP1920-600 Meadowlark Optics) and was combined with a low-repetition-rate fiber laser for photostimulation (Amplitude Systems, Satsuma, pulse duration 300 fs, repetition rate 500 kHz, wavelength $\lambda = 1030$ nm, maximum power 10 W).

The LC-SLM was divided in 45 horizontal tiles, each independently configurable. Each tiled hologram could be encoded with different sets of 3D diffraction-limited spots enabling to multiplex the temporally focused gaussian beam in multiple targeted locations on the sample. The phase profile of each zone was independently calculated using adapted implementations of weighted Gerchberg and Saxton algorithm[23,56–58] in Python. A beam stop was placed on an intermediate imaging plane to physically block the SLM's not modulated zero order. Each tile of the LC-SLM was illuminated by deflecting the GM of a certain angle, corresponding to a precise driven voltage. A calibration was done in order to associate the beam position on the LC-SLM and the voltage to be applied on the GM. Additionally, fast modulation by the AOM allowed to adapt laser power accordingly with the number of spots in each hologram, their diffraction efficiency and the efficiency of each horizontal tiles.

During *hybrid*-FLiT experiments for sub-ms desynchronization of pairs of neurons, the AOM and GMs was driven with a Digidata 1440A interface and pClamp software (Molecular Devices). In *hybrid*-FLiT experiments for mimicking of random spike patterns and *cyclic*-FLiT experiments, the system was controlled with a digital-analog converter board (National Instrument, USB-6259). The control of the system was fully automatized through homemade software written in Matlab and Python 3 (using the open graphic library PyQt5) which allowed the control of the SLM, the GM rotation and the AOM attenuation.

For *cyclic*-FLiT in Fig. 4 and Supplementary Figs. 11–17, two different configurations were used, named configuration S4-a and S4-b (Supplementary Fig. 12), using 23 or 16 lines on the SLM and providing an axial resolution of $\approx 26$ or $\approx 17$ µm across $0.35 \times 0.35 \times 1$ mm³, respectively.

### Optical characterization of two-photon excitation

In order to optically characterize the system performance, 2P holographic fluorescence patterns were collected by exciting a thin (~1 µm) spin-coated layer of rhodamine-6G in polymethyl methacrylate 2% w/v in chloroform.

Holographic patterns were projected on the sample plane through an excitation objective Olympus LUMPlanFL 60XW NA 0.9 (for Supplementary Fig. 3) or Zeiss W Plan-Apochromat 20× NA 1.0 (for Supplementary Figs. 4, 11, 12). Images were collected by an opposite

imaging objective Olympus LUMPlanFL 60XW NA 0.7 (for Supplementary Figs. 3, 11) in transmission geometry and detected by a CCD camera (pco, panda 4.2 bi or Hamamatsu ORCA-05G). A short-pass filter rejected laser light (Chroma Technology 640DCSPXR; Semrock, Brightline Multiphot on Filter 680/sp). 3D stacks were collected by maintaining the excitation objective in a fixed position and moving the imaging objective along $z$ direction with 1-µm steps by means of a piezoelectric motor.

Axial distribution of intensity on different spots was measured by integrating the pixel intensity across circular region of observations (ROIs) around the spots in each z plane. Each axial intensity distribution was fitted with a Lorentzian model. The intensity and axial resolution for each spot was evaluated and reported as maximum intensity and FWHM of the fitted curves, respectively. Images were analyzed with ImageJ and 3D rendering was performed by Imaris. In Supplementary Fig. 3e, f, axial resolution of in-focus spots was measured by averaging the axial resolution of individual spots distributed in a two-dimensional $5 \times 5$ spots matrix in the field of excitation of each tiled hologram (30 µm inter-spots distance). In Supplementary Fig. 3c, d, in-focus intensity homogeneity of each FOV was measured by generating two-dimensional groups of 10 spots randomly distributed in the FOV of each tiled hologram. In Supplementary Fig. 3j, the axial resolution of spots distributed in a 3D volume was obtained by averaging the axial resolution of groups of eight spots randomly distributed in a $120 \times 120 \times 300$ µm for each tiled hologram. In Supplementary Fig. 11, the axial resolution of spots distributed in a 3D volume was obtained by averaging the axial resolution of groups of 8 spots randomly distributed in a $0.35 \times 0.35 \times 1$ mm³ for each tiled hologram.

Notably, tiled holograms are here only used to multiplex a temporally focused Gaussian spot but not to control its shape, this enabled to tile the SLM with multiple (up to 45) holograms without deterioration of spot quality (Supplementary Movie 2 and Supplementary Fig. 11); this is opposite to the quick spot deterioration when small holograms are used to also control the spot shape (Supplementary Fig. 13).

### Characterization of the switching time between tiles of the LC-SLM

We characterized the switching time to reposition the beam on different tiles of the LC-SLM by means of a photodiode as schematized in Figs. 2 and 3.

For the data shown in Fig. 2b–d, we measured the time needed to switch between adjacent tile $i$ and tile $i+1$ of the LC-SLM subdivided in 20 holograms. This configuration is necessary when the illumination time for the different hologram addressed on the LC-SLM must be set independently. For example, if we want to impose a spiking delay of 0.5 ms, among two neurons A and B (similarly to what described in Fig. 2), knowing that neuron A spikes when illuminated for a duration e.g., $t_A = 5$ ms and neuron B spikes when illuminated for a duration e.g., $t_B = 6$ ms, we need to illuminate the first hologram 0.5 ms, the second for 4.5 ms and the third for 1.5 ms. For this, the galvo mirror must switch between three different tilt angles at the fastest possible speed, and steadily remain on each position for the corresponding hologram dwell time (i.e., 0.5, 4.5, and 1.5, respectively). This means that the galvo from one static position needs to rotate at its maximum speed towards a second static position. To reach the maximum velocity, the galvo needs then to be driven with a step voltage waveform from voltage $V_i$ to $V_{i+1}$, where $V_i$ is the voltage corresponding to the galvo at a certain angle $\alpha_i$. To follow the command, the galvo firstly maximally accelerate and then decelerate to accurately reach the targeted position. For our system, the step response for the small-angle deflection we need is of 90 µs (in agreement with small-angle step response of 100 µs given from the manufacturer).

In order to test the timing of this configuration, we generated two distinct phase masks, $\varphi_i$ and $\varphi_{i+1}$, each encoding for an individual spot

placed in a specific XY location of the focal plane as depicted in Fig. 2b. We positioned the photodiode (PD) in a conjugated plane of the sample and we aligned it such that the spot illuminates the center of the detector. We displayed $\varphi_i$ on the tile $i$ and we recorded the light intensity on the PD, while driving the GM servo with a single-step voltage pulse (pulse width 1 s) which deflect the beam across small angles between tile $i$ to tile $i+1$. We repeated the same procedure by displaying $\varphi_{i+1}$ on tile $i+1$ (Fig. 2c). From these two measurements, we obtained the averaged switching time to move between two consecutive tiled holograms in opposite directions, as the time taken for the signal to rise/fall between 3% and 97% of the maximum intensity (Fig. 2d). Of note, the position of PD was finely adjusted to maximize the photon counting when the GM was stationary positioned on tile $i$ or tile $i+1$.

For the data shown in Fig. 3c–e, we measured the minimal switching time between holograms when sequentially scanning at constant rate all holograms. This configuration is necessary when different holograms need to be sequentially addressed at a constant illumination time per target as in *cyclic*-FLiT. That is the case when light needs to be quickly repositioned in series on multiple locations. In this case, the galvo can be driven with a different command waveform. We experimentally verified that a good compromise between speed and constant rate was obtained when the galvo was driven with 50 μs step-duration staircase function (where each step-jump amplitude corresponds to the deflection angle to address different hologram on the SLM) (Fig. 3d). We generated a hologram $\varphi_i$ on a single tile $i$ encoding for an individual spot detected by the PD as previously described. We then recorded the light intensity on the PD, while driving the GM servo with a staircase voltage pulse (pulse time interval 50 μs), which deflects the beam across wide angles between tile 1 to tile 20. From that, we measured the beam dwell time on hologram $\varphi_i$ during switch between hologram $\varphi_1$ and hologram $\varphi_{20}$. We repeated the same procedure for all 20 holograms. From that, we measured the beam dwell time on each tiled hologram as the FWHM of the illuminating pulse duration during whole scan of all holograms at constant switch rate (Fig. 3e). Of note, scan of all holograms would be alternatively possible by driving the GM with a single-step voltage facilitating maximum speed deflection of the beam across wide angles between tile 1 to tile 20. While that can facilitate shorter dwell time per hologram, it also gives variable dwell time per holograms as central tiled holograms feature shorter illumination dwell-times compared to distal tiled holograms as mirror reaches maximum speed at the midpoint.

For the results shown in Supplementary Fig. 4, we sequentially switched between four illumination patterns, obtained either by using galvos switch in FLiT configuration or using the conventional SLM refreshing process. We used a HSP1920-600 Meadowlark Optics Spatial Light Modulator, characterized by nominal fast response rate ≈30 Hz. The fluorescence signals were detected using a DaVinci 2K (RedShirtImaging) fast CMOS camera, operated at 5 kHz. For a higher temporal resolution, a photodiode (Thorlabs DET36A/M, 14 ns rise time) was used to detect the laser intensity (Supplementary Fig. 4d). Camera detection, galvo movement and SLM pattern-switch were controlled and synchronized with a National Instruments acquisition card.

## Animals

All procedures involving animals were in accordance with national and European (2010/63/EU) guidelines and were approved by the authors' institutional review boards and national authorities (French Ministry of Research, protocol ID: 02230.02). Experiments were performed on C57BL/6J male and female mice (Jackson lab.) and transgenic fish lines (Danio rerio) obtained by injecting a 14UAS:CoChR-Tdtomato plasmid in a background expressing Gal4 under the *Vglut2a* promoter (tg(Vglut2a:gal4), targeting glutamatergic neurons) and GCaMP6s under the pan-neuronal *HuC* promoter (tg(HuC:H2B-GCaMP6s). Mice

were reared in a 12 h light/dark cycle with food ad libitum and fish were maintained at 28 °C on a 14 h light/10 h dark cycle. Animals were housed in the animal facilities of the Vision Institute, which were built according to the respective local animal welfare standards. All efforts were made to minimize suffering and reduce the number of animals.

## In vivo viral expression

Stereotaxic injections of the fast somatic opsin ST-ChroME were performed in 3-week-old male mice. Mice were anesthetized with ketamine (80 mg/kg)–xylazine (5 mg/kg) solution and a small craniotomy (0.7 mm) was made on the skull overlying V1 cortex. Injection of 1 μl of the solution containing the viral vector was made with a cannula at a rate of 80–100 nl/min and 200–250 μm below the dural surface. We used a viral mixture containing the somatic opsin ST-ChroME (AAV9-hSyn-DIO-ChroME-Flag-ST-P2A-H2B-mRuby-WPRE-SV40, from the Adesnik lab, Berkeley, viral titer of $5.86 \times 10^{13}$ particles/ml) and the Cre recombinase (AAV9-hSyn-Cre, from Addgene, $3.3 \times 10^{13}$ p/ml), diluted at a factor 10 and 100, respectively, in fresh NaCl solution. The craniotomy and the skull were then sutured and the mouse recovered from anesthesia. After 2–3 weeks, sufficient for an adequate expression of the virus, mice were used for electrophysiological experiments. ST-ChroME expression in acute cortical slices is shown in Supplementary Fig. 5.

For experiments of Supplementary Fig. 16a–c, stereotaxic injections were performed to deliver high titer (in the order of $10^{12}$ vp/ml) of adeno-associated viral vectors AAV9-syn-jGCaMP7s-WPRE (Addgene) and AAV1-hSyn-ChRmine-mscarlet-Kv2.1-WPRE (plasmid from Addgene), at a ratio of 2:1, in the barrel cortex of 5–8 weeks-old mice. Mice were anesthetized with intraperitoneal injection of a mixture of ketamine (60-80 mg/Kg)–xylazine (5–8 mg/Kg), and subcutaneous injections of analgesics, buprenorphine (0.1 mg/Kg) and lidocaïne. A craniotomy of 0.5–0.7 mm was made on the skull overlying wS1 cortex (−1.5 mm from Bregma/2.5 mm lateral/100–300 μm deep), and 1 μl solution containing the viral vector was delivered. Three to five weeks after the viral injection, the cranial window surgery was performed under ketamine/xylazine. A cranial window of 3–3.5 mm diameter was made above and centered on the viral injection site. Dura was removed while keeping the brain surface moist with sterile saline solution. A 3-mm cover glass (#0, 0.085 to 0.13 mm thickness; Multichannel System) was placed on top of the craniotomy and sealed with dental cement. A head plate was then fixed with dental cement. Mice were given at least 3 days to recover from the cranial window surgery.

## Preparation of organotypic cultures and viral infection

Hippocampal slices cultures were prepared from postnatal day 6–9 mice pups according to the interface culture method[59]. Briefly, hippocampi were gently detached from the brain and placed in a cold dissecting medium composed of: Gey's Balanced Salt Solution (Sigma G9779), supplemented with 25 mM D-glucose, 10 mM HEPES, 1 mM Na-Pyruvate, 0.5 mM α-tocopherol, 20 nM ascorbic acid and 0.4% penicillin/streptomycin (5000 U/mL; Fisher 11528876). Transverse slices of 300-μm thickness were cut using a McIlwain tissue chopper, maintained for at least 1 h at 4 °C and then transferred onto semiporous membranes inserts (47 mm diameter, 0.45-μm pore size; Millipore FHLP04700) which were placed in six well tissue culture plates containing 1.1 ml medium per well. The incubation medium consisted in: 50% Opti-MEM (Fisher 15392402), 25% heat-inactivated horse serum (Fisher 10368902), 24% HBSS (Fisher 15266355), 1% penicillin/streptomycin (5000 U/mL), and supplemented with 25 mM D-glucose, 1 mM Na-Pyruvate, 20 nM ascorbic acid and 0.5 mM α-tocopherol. Slices were maintained at 34 °C in an incubator with 5% $CO_2$. After 3 days, the medium was replaced with a fresh and warm Neurobasal culture medium composed of: 2% Neurobasal-A (Fisher 11570426), 15% heat-inactivated horse serum, 2% B27 supplement (Fisher 11530536), 1% penicillin/streptomycin (5000 U/mL), and supplemented with 0.8 mM

L-glutamine, 0.8 mM Na-Pyruvate, 10 nM ascorbic acid and 0.5 mM α-tocopherol. This medium was changed every 2–3 days until the experiment.

Organotypic slices were then infected with 1 µL of virus at 5 days in vitro (DIV). We used the same mixture as for in vivo stereotaxic injections. Slices were used for electrophysiology recordings at 12–14 DIV. See Supplementary Fig. 5 for ST-ChroME expression in this preparation. For all-optical experiments, slices were also infected with the virus Gcamp7s (AAV9-Syn-jGCaMP7s-WPRE, from Addgene, viral titer of $2.3 \times 10^{13}$ particles/ml) at 4 days before the experiment. Supplementary Fig. 5 shows the co-expression of ST-ChroME opsin and the calcium indicator GCaMP7s.

### Acute slice preparation for electrophysiology

Acute parasagittal slices of the visual cortex were prepared from adult mice 2–3 weeks after viral injection. Animals were decapitated after being deeply anesthetized with isoflurane (5% in air). The brain was quickly removed, immersed in an ice-cold choline solution and 300 µm thickness slices were obtained using a vibratome (Leica Biosystems VT1200S). The cutting solution contained the following (in mM): 126 choline chloride, 16 glucose, 26 NaHCO$_3$, 2.5 KCl, 1.25 NaH$_2$PO$_4$, 7 MgSO$_4$, 0.5 CaCl$_2$, pH 7.4, cooled to 4 °C and equilibrated with 95% O$_2$/5% CO$_2$. Slices were maintained at 32 °C for 20 min in standard ACSF (sACSF) containing the following (in mM): 125 NaCl, 2.5 KCl, 26 NaHCO$_3$, 1.25 NaH$_2$PO$_4$, 1 MgCl$_2$, 1.5 CaCl$_2$, 25 glucose, and 0.5 ascorbic acid, pH 7.4, saturated with 95% O2 and 5% CO$_2$ and then transferred at room temperature in the same solution until recordings.

### In vitro whole-cell electrophysiology

Acute slices as well as organotypic slices were placed in a recording chamber under the microscope objective, and perfused continuously with fresh sACSF saturated with 95% O$_2$ and 5% CO$_2$. Neurons were patched at 30–60 µm from the slice surface. Single or double-patched neurons were clamped at −70 mV in voltage-clamp configuration and membrane potential was kept at −70 mV with currents injections in current-clamp configuration. Patch electrodes (Borosilicate glass pipette, outer diameter 1.5 mm and inner diameter 0.86 mm, Sutter Instruments) were filled with an intracellular solution containing the following (in mM): 127 K-gluconate, 6 KCl, 10 Hepes, 1 EGTA, 2 MgCl2, 4 Mg-ATP, 0.3 Na-GTP; pH adjusted to 7.4 with KOH. The estimated reversal potential for chloride (E$_{Cl}$) was approximately −69 mV based on the Nernst equation. Pipettes were pulled from borosilicate glass capillaries and had a typical tip resistance of 5–6 MΩ. The averaged serie resistances were 18.5 ± 7.9 MΩ ($n = 34$ cells) and 17.9 ± 3.7 MΩ ($n = 8$ cells), for acute slices and organotypic cultures, respectively. The following receptor blockers were added to the sACSF to block any synaptic effect: DNQX and AP-V (1 µM each; from Abcam). Electrophysiology data were acquired with a Multiclamp 700B amplifier and digitized with a Digidata 1322 A interface and pClamp software (Molecular Devices). Signals were sampled at 20–50 kHz and filtered at 4–10 kHz.

### Desynchronization of activity of distinct neurons

In *hybrid*-FLiT experiments shown in Fig. 2, we desynchronized two ST-ChroME-expressing targeted neurons, here called neuron A and neuron B. The following photostimulation procedure was used in order to trigger activity in neuron A and B with a time delay shorter than the illumination dwell-times needed to evoke activity in the two neurons. We defined three tiled phase masks and we vertically piled them adjacently on the LC-SLM display such that: tile φ$_A$ encodes for illumination of neuron A (top tile), tile φ$_{AB}$ encodes for simultaneous illumination of neurons A and B (middle tile), and tile φ$_B$ encodes for illumination of neurons B (bottom tile). First, we established threshold light powers $P_A$ and $P_B$, and illumination dwell-times $t_A$, $t_B$ to independently evoke an AP on neuron A and neuron B, by deflecting the GM on

tile φ$_A$ and φ$_B$. Threshold values were defined in current-clamp mode when AP was reliably generated on 3/3 consecutive trials (40 s inter-time between trials). Photocurrents corresponding to threshold illumination conditions were also recorded in voltage-clamp. On the basis of these values, we set a sequence to drive the GM and the AOM and introduce arbitrarily defined spike delays δt between neuron A and B. Accordingly, the beam was sequentially directed by tilting the GM on tile φ$_A$ for a time δt, on tile φ$_{AB}$ for a time $t_A − δt$ and on tile φ$_B$ for a time $t_B − (t_A − δt)$. The incoming power was adjusted via the AOM such that it was set to $P_A$ when the beam was on tile φ$_A$, to $P_B$ when the beam was on tile φ$_B$ and to $P_A + P_B$ when the beam was on tile φ$_{AB}$. Of note, the diffraction efficiency of phase mask φ$_{AB}$ was computationally corrected such that the ratio of intensity sent onto neuron A and B equals to $P_A/P_B$. That ensures that both neurons are constantly illuminated with the same intensity during their respective illumination dwell time. Importantly, in voltage-clamp mode, we verified that the beam of power $P_A + P_B$ positioned on φ$_{AB}$ for a time $t_A$ and $t_B$ elicited the same photocurrents previously elicited by illuminating only neuron A (with $P_A$ power, $t_A$ dwell time on φ$_A$) and B (with $P_B$ power, $t_B$ dwell time on φ$_B$), respectively (Supplementary Fig. 7e, f). GM deflection between the three tiles was driven with small-angle single-step voltage as previously detailed. We thus recorded in current clamp the APs driven in neuron A and B by addressing GM and AOM following the established sequence of photoactivation.

Notably, for all those experiments which feature delays longer than the illumination dwell-times needed to evoke activity in the two neurons, only two tiles of the LC-SLM are necessary (tile φ$_A$ and tile φ$_B$), as the beam will never be simultaneously on neuron A and B.

Supplementary Fig. 2 schematizes how this strategy can be generalized to desynchronize *n* neurons (or n groups of neurons) with delays inferior to each activation dwell time, by dividing the LC-SLM in $2n − 1$ tiled holograms and piling them on the LC-SLM such that each hologram encodes, from top to bottom: 1st tiled hologram → 1st neuron, 2nd tiled hologram → 1st + 2nd neurons,…, *n*th tiled hologram → 1st + 2nd +…+*n*th neurons, *(N + 1)*th tiled hologram → 2nd+…+*n*th neurons, *(n + 2)*th tiled hologram →3rd+…+*n*th neurons … *(2n − 1)*th tiled hologram → *n*th neuron. Power on each of the $2n − 1$ tiled phase masks needs to be modulated accordingly to the number of encoded targets.

In *hybrid*-FLiT experiments aiming to mimic neuronal firing shown in Fig. 2e and Supplementary Fig. 9, reference traces originated from an individual recording under in vivo patch clamp. In particular, two subsections, each 2 s long and featuring characteristic firing patterns, were arbitrarily selected and delayed. The two traces were then feed to a homemade software which extracted the spike timing and automatically determined the illumination sequence (including illumination power and switch time) to be addressed on the tiled holograms of the LC-SLM to reproduce the delayed spiking patterns on two double-patched neurons.

### Multi-neuron activation with electrophysiological recording

In *cyclic*-FLiT experiments shown in Fig. 3, the LC-SLM was subdivided in *n* tiled phase masks. In particular, one mask φ was encoded to illuminate one targeted ST-ChroME-positive neuron. We initially established threshold light power $P_{std}$ and illumination dwell time $t_{dw}$ to evoke an AP on the cell by tilting the GM to steadily illuminate φ.

The cell was then photoactivated under *cyclic*-illumination by driving the GM with a staircase voltage input which facilitated steering the beam back and forth on all *n* holograms through discrete angle deflections and fixed dwell time per hologram at $t_{cyc} = 50 \,\mu s$ for $N_{cyc}$ cycles (Supplementary Fig. 19).

We tested a slow photoactivation protocol featuring a total scan time equals to $n \cdot N_{cyc} \cdot t_{cyc}$ (Supplementary Fig. 10) and a fast photoactivation protocol featuring a total scan time equals to $t_{std}$ (*hybrid*-FLiT, Fig. 3f, g). We established power to trigger an AP in both cases by

varying the number of the tiled holograms (i.e., the size of each tiled hologram) between 12 and 50. Photocurrents have been recorded in voltage-clamp by displaying $\varphi$ on different position of the LC-SLM, in order to verify that different tiles substantially elicit the same photocurrent (Supplementary Fig. 7e, f).

For experiments in Supplementary Fig. 15, power curves and axial resolutions of the photostimulation are obtained under single-cell patch-clamp recordings in organotypic slices during steady illumination ($H = 1$) with power $P_{std}$ or a cyclic scan with power $P_{cyc} = P_{std}\sqrt{H}$ over $H = 16$ or 23 holograms, one of which targeting the patched cell. Axial resolution curves are obtained by moving axially the microscope objective to shift the illumination spots along the $z$ axis while recording photoinduced currents or APs. For each condition, AP probability is estimated over ten stimulation pulses (10 ms each) at 10 Hz.

### Multi-neuron activation with functional imaging read-out

In *cyclic*-FLiT all-optical experiments reported in Fig. 4 and Supplementary Figs. 11–17, the LC-SLM was subdivided in $H = 45$ tiled holograms. A total number of 23 cells (6 FOVs) or 69 cells (1 FOV) per FOV distributed in 2D (4 FOVs, $365 \times 365\,\mu m^2$) or in 3D (3 FOVs; $365 \times 365 \times 60\,\mu m^3$) were targeted. Cells were photoactivated under steady illumination by encoding all cells in the central hologram and under *cyclic*-illumination by partitioning the cells among the 4, 9, 16, or 23 central holograms. In conventional steady illumination, cells were photostimulated with 10 pulses of 10 ms duration at 10 Hz. In *cyclic*-illumination, the GM was driven with a staircase voltage input which facilitated steering the beam back and forth on 4, 9, 16, or 23 holograms with fixed dwell time per hologram at $t_{cyc} = 50\,\mu s$ for 10 ms. This *cyclic*-illumination was repeated for ten times at 10 Hz, such that the same illumination protocol performed in steady illumination was reproduced in a cyclic manner. This protocol was repeated for different local illumination powers sent to each spot ($P_{spot}$): 5 mW, 7.5 mW, 10 mW, 15 mW, 20 mW, 25 mW, 35 mW per cell in conventional holography and 5 mW·$\sqrt{H}$, 7.5 mW·$\sqrt{H}$, 10 mW·$\sqrt{H}$, 15 mW·$\sqrt{H}$, 20 mW, 25 mW·$\sqrt{H}$, 35 mW·$\sqrt{H}$ per cell in *cyclic*-illumination. The reported power per cell (*P/cell*) on the axis of Fig. 4 and Supplementary Fig. 16 corresponds to the total power sent to the sample divided by the total number of targets.

To achieve uniform light distribution in the FOV and among the multiple holograms, the illumination power was adjusted to compensate the position-dependent diffraction efficiency and power losses due to the different positions of the tiled holograms at the objective aperture. The first effect was compensated by using diffraction efficient corrected holograms[23,58], the second by using an AOM which rapidly adjust the laser power during the scanning of the tiles. This second correction depends on the size of the SLM at the objective back aperture and the magnitude of the grating induced dispersion. The choice of these parameters is a trade off between achievable FOV, axial resolution and power losses as in conventional MTF-LS. In all-optical experiments, we chose an average vertical size of the illumination spot on the LC-SLM as a compromise that slightly underfilled the holograms for low value of $H$ and overfilled the holograms in the case of large $H$, with consequent losses of intensity in the latter case. Once the optimal number of holograms is chosen this value can be optimized to better match the size of $H$ and minimize power losses (Supplementary Fig. 12). The axial resolution achievable is a compromise among power losses and FOV, and can be adjusted by varying the illumination of the SLM and OBA. We used two configurations (S4-a and S4-b) where we reached an axial resolution between $17.5 \pm 4.5\,\mu m$ and $26 \pm 8\,\mu m$ (Supplementary Figs. 11 and 12), in agreement with previously reported value using similar objectives[10,11,20,29,36]. In Supplementary Fig. 12c, power losses vs number of holograms have been calculated from the reduction of the fluorescence induced by a set of holographic spots exciting a Rhodamine-6G thin layer. The FLiT throughput efficiency $K$ reported in Supplementary Fig. 12d–f has been calculated as $K = \frac{n_{cyc,\,exp}}{n_{std}}$, where $n_{std} = m \cdot H$ *is* the number of cells activable using a light power $P_{holo}$ in conventional illumination and $n_{cyc,\,exp} = \left\langle m_{cyc,\,exp} \right\rangle \cdot H$ is the number of cells activable using a light power $P_{cyc} = P_{holo}\sqrt{H}$ in *cyclic*-illumination, with $\left\langle m_{cyc,\,exp} \right\rangle = \frac{1}{H}\sum_{h=1}^{H} \frac{1}{C_f(h)} \cdot m$ the average number of cells activable per hologram given the experimental power correction function $C_f(h)$.

Imaging was performed with a 920 nm beam (Ti:Sapph, Mai-Tai Spectra Physics) going through a commercial galvo-galvo scanning head (VIVO 2-PHOTON operated using Slidebook 6 software, 3i) and achieving a typical 2D raster-scanning in $365 \times 365\,\mu m$ FOV or multifoci scanning 3D in $365 \times 365 \times 60\,\mu m$ at 3.15 Hz. For 3D multifoci imaging, a phase modulation of the 920 nm beam produced by an SLM (Hamamatsu X13138-07) placed in an upstream conjugate plane of the $x$, $y$ scanning galvanometric mirrors (as depicted in Fig. 4a and similar to[60]) allowed the generation of a multifoci point spread function and the simultaneous scanning of three planes of the sample at typically $+30\,\mu m$, $0\,\mu m$, and $-30\,\mu m$ from the objective focal plane. Fluorescence was simultaneously detected across the three planes and axially projected on a single 2D $365 \times 365\,\mu m$ image. Imaging power ranged typically between 8 and 20 mW per plane depending on the GCaMP7s expression level, which is expected to give an artefactual depolarization of $\leq 5\,mV$ and no artefactual spiking generation with ChroME and GCaMP[61]. The system was used to collect data reported in Fig. 4 and Supplementary Figs. 4,11–17.

For in vivo experiments, mice were maintained head-fixed under isoflurane anesthesia (<1.5%) for 2–3 h, while body temperature and anesthesia parameters were monitored along the experiment (homeothermic system, Kent Scientific). Subcutaneous injections of 0.1 ml sterile saline solution ensured rehydration all along the experiment. Simultaneous photostimulation (under *cyclic*-FLiT or conventional holography) and GcaMP7s imaging was performed. At least 3 days of recovery were given between successive acquisition days for every animal, and cranial windows remained clear for more than a month.

For in vivo experiments involving zebrafish, 4–5 days post fertilization (dpf) larvae were mounted in 2.5% low melting-point agarose in a 35-mm Petri dish. After a period of acclimatization, larvae were placed under the microscope and simultaneous photostimulation (under *cyclic*-FLiT or conventional holography) and imaging of GcaMP6s transients was performed. The same lateral spot size (FWHM≈13 µm) was used on experiments targeting mammalians neurons or zebrafish cells.

### Analysis of functional imaging

In the results shown in Fig. 4 and Supplementary Figs. 14,16,17, GCaMP fluorescent was integrated over circular ROIs (diameter between 13 and 15 µm for mammalians neurons, and 8 µm for zebrafish neurons) centered on individual targeted somata. Photostimulation artifacts were semi-automatically detected and removed by a GUI-based Matlab script. Percentage changes of fluorescence were computed as $\frac{\Delta F}{F} = \frac{F - F_0}{F}$, where $F_0$ is the basal level of fluorescence measured averaging 15 frames (-5 s) before each stimulation event. Under multiplane imaging, potential $x$–$y$ overlaps of axially aligned somata were identified and the $\Delta F/F$ was extracted only from non-overlapping regions. A cell was considered responding if $\frac{\Delta F}{F} > 0.2$. In in vivo recordings, photostimulation trials presenting large spontaneous activity were discarded.

### Photodamage experiments on GCaMP7s organotypic slices

In Supplementary Fig. 17, we used the increment of fluorescence as an indicator of the cell health during prolonged illumination protocols at different powers. We measured $F/F0$, with $F0$ the integrated fluorescence intensity for each target cell at the beginning of the recording. In

Supplementary Fig. 17a, b, each stimulation trials (130 s) consisted of multiple stimulation events, each consisting of a train of 10 light pulses (pulses of 10 ms pulses at 10 Hz) under *cyclic* and conventional illumination at increasing powers $P_{cyc}$ and $P_{std}$ during GCaMP imaging. We recorded multiple trials at increasing powers. The *F/F*0 traces were then extracted for each target cell and concatenated after removal of stimulation artifacts. The total number of simultaneously targeted cell per experiment was varying between 4 and 23. In Supplementary Fig. 17c, we used longer *cyclic* stimulation protocols (GCaMP imaging for 15 s followed by photostimulation train of 10 ms pulses at 10 Hz for a total time of 5 min, repeated five times). In total, 23 cells were targeted using $H = 23$ holograms and illuminated with $P_{spot} = 15\,\text{mW}\cdot\sqrt{H} = 72\,\text{mW}$ (as used in Fig. 4g). The induced *F/F*0 fluorescence increase was evaluated as the ratio between the fluorescent intensity of each target cell before and after each trial.

### Temperature simulation

The spatiotemporal distribution of the temperature rise reported in Fig. 5 was calculated by solving the Fourier heat diffusion equation[62] considering the brain tissue as an infinite medium with isotropic and uniform thermal properties as described in ref. [25]. The solution is obtained by convolving the Green's function for the diffusion equation by the thermal source term, which is the thermalization of the absorbed light source intensity. This model has been experimentally validated[25]. Numerical solution was implemented in Python, taking special care in selection spatial and temporal sampling to avoid overlap (aliasing) due to cyclic boundary conditions induced by the use of Fourier transform-based numerical convolution. Time-averaged temperature rise for *cyclic*-FLiT in Fig. 5 is a rolling averaged during the illumination period (10 ms). In order to compensate for scattering, the power sent to a cell at a specific depth z below the surface is increased by a factor: $\chi = e^{z/\ell_s}$, where $\ell_s$ is the scattering length (in biological tissue, $\ell_s = 166\,\mu\text{m}$ for a wavelength of $1.03\,\mu\text{m}$[63]. Considering cells distributed uniformly in depth between $z_1$ and $z_2$, we can calculate the average value of this factor $\langle\chi\rangle = \frac{1}{\Delta z}\int_{z_1}^{z_2}\chi dz = \frac{\ell_s}{\Delta z}e^{z_1/\ell_s}\left(e^{\Delta z/\ell_s} - 1\right)$ where $\Delta z = z_1 - z_2$.

### Data analysis and statistical tests

We performed the analysis of the recorded stacks on Rhodamine layers with MATLAB, ImageJ, and the Imaris software (Bitplane, Oxford Instruments). The 2P fluorescence values for each spot were obtained by integrating the intensity of all the pixels in a circular area containing the spot, in the plane where the intensity was at its maximum value (i.e., the TF plane). Axial intensity distributions were obtained by integrating the intensity of the pixels in the same area for each plane of the recorded stack, in a range of $\pm 20\,\mu\text{m}$ around the focal plane of each spot. Reported values for the axial confinement were the fit of the axial profile of the spots with a Lorentzian model and referred to the FWHM of the curves.

All electrophysiological data were analyzed with Clampfit (Molecular Devices) and Origin Lab Pro. For *hybrid*-FLiT experiments in Fig. 2, we measured, for double-patched neurons, A and B, the depolarization onset or the AP peak delay, determined as the time between the beginning of the light stimulus and membrane potential change or the AP peak, respectively. We then subtracted the values of cell B to cell A and compared this temporal delay to the expected one. We evaluated the temporal accuracy as the difference between imposed δt and experimental $\delta t_{AP}^{exp}$ delays, $|\delta t_{AP}^{exp} - \delta t|$. Global accuracy was calculated as weighted mean and SD of all imposed δt. For mimicking experiments under *hybrid*-illumination, the analysis of the results was established by pairing the closest subsequent APs in the two neurons. In particular, for each AP pair, we evaluated the temporal accuracy as the difference between driven and experimental inter-spike time. We then calculated the overall accuracy of the mimicking by weight averaging the temporal accuracy of each AP pair.

AP latencies were determined as the time between the beginning of the stimulus and the time of AP peak, and AP jitters were calculated as the standard deviation (SD) of the AP latency across trials. All recordings were analyzed and averaged across 3–5 photostimulation trials. All statistical analyses were performed in Prism (GraphPad Software, Inc.). Normality of the data was systematically assessed (D'Agostino and Pearson omnibus normality test). We performed non-parametric tests as distributions were not normal or *n* was small (two-tailed Mann–Whitney test or Kruskal–Wallis ANOVA test followed by Dunn's multiple comparison test for more than two groups). Differences were considered non-significant (ns) if $P > 0.05$. All values are presented as mean ± SD of *n* experiments except when differently specified.

### Reporting summary

Further information on research design is available in the Nature Portfolio Reporting Summary linked to this article.

### Data availability

Access to additional datasets will be provided upon request to the corresponding authors. Source data are provided with this paper.

### Code availability

Phase hologram calculations are based on published algorithms[23,56–58]. The codes to control the galvanometric system, the SLM addressing, and to define illumination parameters for FLiT stimulation are publicly available in the GitHub repository: https://github.com/photonics-VisionInstitute/FLiT.

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

## Acknowledgements

We thank Florence Bui and Valeria Zampini for helping with stereotaxic injections and Imane Bendifallah with organotypic cultures. We thank Nicolò Accanto for the helpful discussion on the methods of illumination. We thank Ruth Sims for proofreading of the manuscript and Hillel Adesnik for providing the opsin ST-ChroME. We thank the IHU FOReSIGHT (Grant P-ALLOP3-IHU-000), the Fondation Bettencourt Schueller (Prix Coups d'élan pour la recherche française to V.E.), the National Institute of Health (Grant NIH 1UF1NS107574–01 to V.E.), the Institut Carnot Voir et Entendre (GrantP-SPAM00-ANR-000 to D.T.), the Axa research funding, (Axa Chair to V.E.) the Région Île-de-France (Grant WASCO; DIM-Elicit to V.E.) and the European Research Council (HOLOVIS; ERC2019-ADG-885090 to V.E.) for financial support.

## Author contributions

C.M., C.T., and E.R. built up and characterized the high magnification setup. M.H., C.Te., G.F., and D.T. built up and characterized the low magnification setup. C.M., C.Te., D.T., and V.d.S. developed the interface for fast holographic switching with the implementation of hologram calculations. G.F. performed the electrophysiological recordings with the contribution of C.M. G.F., M.H., and D.T. performed all-optical experiments and photodamage threshold tests in vitro. F.B. and D.T. performed in vivo all-optical experiments in mice. G.F. and D.T. performed in vivo all-optical experiments on zebrafish larvae. G.F., C.M., C.Te., M.H., D.T., and E.R. performed data analysis. B.C.F., E.R., and V.E. designed and performed thermal and opsin photocycle simulations. G.F. and F.D.B. generated and provided zebrafish lines for in vivo experiments. C.T. provided instrumental resources. E.R. and V.E. wrote the original draft. E.R. and V.E. reviewed and edited the final manuscript with contributions from G.F., D.T., and B.C.F. E.R. and V.E. conceived and supervised the project.

## Competing interests

The authors declare no competing interests.
