## [Peer Review File · Nature Communications]

Ultrafast Light Targeting for High-Throughput Precise Control of Neuronal NetworksThis manuscript has been previously reviewed at another journal that is not operating a transparent peer review scheme. This document only contains reviewer comments and rebuttal letters for versions considered at *Nature Communications*.

REVIEWERS' COMMENTS

Reviewer #3 (Remarks to the Author):

The authors have made significant improvements to the manuscript regarding the details of their protocols and how they relate to stimulation efficiency. It is very helpful to have the additional descriptions of the experiment configurations, as well as the additional modeling exploring the parameter space. I believe this will be a great service to readers to better understand the tradeoffs. With the configurations laid out, it appears that there are important gaps in the exploration of this parameter space, and that key parameters are allowed to vary favorably for cyclic-FliT but not conventional holography. I have attempted to lay out my understanding of the different effects and what the relative difference in efficiencies may more accurately be between conventional holography and cyclic-FliT, below.

Fundamentally, I believe that two methods must be compared to one another using the same experimental parameters whenever possible. In the two main configurations described by the authors in their modeling work, most parameters are held constant but one parameter—time (Configuration 1) or laser power (Configuration 2)—is allowed to increase for cyclic-FliT but not for conventional holography. The authors then take the resulting increase in number of targetable neurons in the cyclic-FliT case to reflect a total improvement in efficiency without accounting for the impact of longer experiment time or increased laser power. In my last review, I had requested that the authors provide modeling and experiments for how much power would be required for conventional holography to match the number of targetable neurons with cyclic-FliT when total experiment time was held constant, or when total dwell time was held constant. In their rebuttal, the authors made some comments about the latter and did not explore the former. When the parameters are matched between methods, or when conventional holography is also permitted to vary parameters for its benefit, I believe that the differences between conventional holography and cyclic-FliT will be found to be far less than what the authors claim throughout the manuscript. This is fundamentally due to the inherent two photon absorption effect governing both methods.

If the relative efficiencies of conventional holography and cyclic FliT are different than claimed by the authors, as I believe they may be, the manuscript must be correspondingly revised. Even if the relative differences are not as pronounced when explored theoretically, there are potential applications for the cyclic-FliT, such as when a strict power ceiling is placed on experiments due to available laser power or damage thresholds. The authors touch on these factors, but the arguments are a bit two-sided and disjointed—arguing there are such limitations but allowing increasing power for cyclic-FliT to relatively high levels per cell and in a supplementary note implying that many more neurons could be targeted with even more power. More tightly describing the power ceiling in the context of practical and biological limitations might point to utility of cyclic-FliT. I also still believe that the hybrid-FliT approach has utility and continue to encourage the authors to focus more on elaborating this approach.

Major comments:

1) Configuration 1 – I do not think it is correct to say that the number of targeted cells is “H times” higher for cyclic FliT in Configuration 1. This statement is off by potentially a large margin once the overall experiment time is taken into account. In Config 1, the cyclic FliT is allowed to increase the experiment time by H fold, but this is not applied to conventional holography in the comparison. An experimenter could easily sequence through multiple holograms using conventional holography over the same longer experiment time, as suggested by the authors, but this is not formally explored. In this case, conventional holography would indeed be limited by the t_{SLM} . The difference in number of targeted neurons will be reduced in sequenced conventional holography by the off duty cycle of the SLM. Since t_{dw} is reported to be 5-10 ms, this off duty cycle (assuming $t_{SLM}=3ms$) would be $3/8$ (37.5%) or $3/13$ (23%). Cyclic-FliT in Config 1 would be theoretically able to fill this fractional off duty cycle time, so would be able to target 1.3 to 1.6 times more neurons in this example $[H/(H*tdw)/(tdw+tSLM)]$, or in simpler form: $1+tSLM/tdw$. This is noteworthy, but much lower than the “H times” more cells stated by the authors. For example, for $H=23$, this is overstating the difference by over 14 times. In practice, achieving ~ 1.6 times more

cells could be achieved within the same dwell time by increasing power by an estimated ~ 1.36 times, incorporating the quadratic relationship of power and excitation. This would likely be a much simpler adjustment for many researchers than implementing cyclic-FlIT.

2) From this, I would argue that in Supplementary Table 1, the Maximum number of cells is:
Conventional Holo: $N_{std_max} = (H*(tdw)/(tdw+tSLM))*(P_{tot}/P')$

3) Configuration 2 – This configuration, where total experiment time is held constant, must be explored in the modeling using parameters where power is allowed to increase for conventional holography. The authors claim that they can target \sqrt{H} more neurons in cyclic-FlIT but use \sqrt{H} more power. They do not increase power for conventional holography by the same amount and ask how many more neurons can be targeted. Given a linear relationship between number of targeted neurons and power in conventional holography (for a fixed dwell time), one could target \sqrt{H} more neurons with \sqrt{H} more power. Indeed, looking at “Maximum Number of Cells” in Configuration 2 Supplementary Table 2, and replacing P_{cyc} with $P_{std}*\sqrt{H}$, as defined in the first row, it appears the \sqrt{H} would cancel out the \sqrt{H} increase for Max number of cells in cyclic-FlIT, wouldn't it? This implies that two methods are equivalent when one accounts for the increase in power. If more power (than \sqrt{H}) is required for conventional holography to target \sqrt{H} more neurons in the same total experiment time, how much more is required and why?

4) For Figure 5, what is the total illumination time for cyclic illumination of a single neuron (How many cycles are repeated for each neuron in cyclic illumination, with what dwell time per cycle)? If that total illumination time per cell in cyclic FlIT is used for conventional holo, and power is increased to $\sqrt{H}*P_{std}$ for the conventional holo case, what does the modeled heating look like? In other words, the nonlinear increase in two photon excitation with higher power is a major factor in reducing heating effects, but this same potential nonlinear benefit is not applied to the conventional holography case for fair comparison.

5) The authors argue that increasing power for conventional holography would lead to damage. Indeed, increasing power will lead to more potential damage. However, this also applies to increasing power in the cyclic-FlIT case, as in the case of using \sqrt{H} more power, with some discussion from the authors in Supplementary Note 2. My understanding is that the experiments typically used $H \sim 20$. $\sqrt{20} = 4.47$. That is a large increase in power. The authors argue that that amount of increase in power is OK for cyclic-FlIT but that it would be damaging to increase power for conventional holography to attempt to match the same number of targeted neurons. The two must be compared on equivalent terms.

6) “However, because of the drastic nonlinear photodamage and thermal effect, this last configuration ($P_{std} \gg P_{sat}$) is very unlikely be adopted in two-photon optogenetics.” This case/assumption should be explored and presented systematically in the text and figures. In the case of cyclic-FlIT, the authors argue that they can use \sqrt{H} more power – e.g., which can be ~ 4.8 times when using 23 holograms, but they do not explore more power for conventional approaches. This should analysis should incorporate nonlinear damage and instantaneous (during illumination) and average (over the same overall experiment time) thermal effects for both methods.

7) The authors state in the response to one of my questions that “As this scheme relies on a single hologram, and all targets are illuminated in parallel, the number of achievable cells with usHolo will be proportionally reduced”. I disagree with this statement. In fact, the number of achievable cells will be equal to the number of cells in the hologram and not reduced. I believe the power used to achieve comparable numbers of neurons as the cyclic-FlIT case (which was more my question) would be the same \sqrt{H} factor, as outlined above.

8) The authors then state that the “maximum number of activatable cells” is proportionally lower with the increase in power, assuming a total power threshold. While technically true, the authors separately argue that several watts of power are available at the objective and also allow cyclic-FlIT to use significantly more power than conventional holo (i.e., \sqrt{H} more power). This use of increased power for cyclic-FlIT will also have limitations regarding the total available power that can be safely used in the experiment.

9) In the rebuttal point "D", Figure R8 does not really address my question, and the limitation of total power applied by the authors may not always be relevant in what they have termed "usHolo". Furthermore, sequential "usHolo" could be implemented to work around total power limitations. How much power would be required when using conventional holography ($H=1$) and the same overall dwell time to stimulate as many neurons as cyclic-FLiT (that uses \sqrt{H} more power for H holograms)? I think this would be $\text{power_per_cell_std} * \sqrt{H} * \text{num_cells}$. For cyclic-FLiT, I believe it is $\text{power_per_cell_std} * \sqrt{H} * \text{num_cells} / H$ indicating the potential advantage. However, in the case of "usHolo", the total illumination would be much briefer: t_std_dw / H compared with overall illumination time in the cyclic case being t_std_dw , a factor of H times longer. Taken over the same overall time t_std_dw the average power would be equivalent. Alternatively, the usHolo approach would permit another hologram to be generated in t_SLM and another set of neurons be targeted potentially sooner than with cyclic-FLiT for $(t_std_dw / H + t_SLM) < t_std_dw$, leading to more overall neurons targeted. The usHolo configuration should be explored and discussed further by the authors and the effects on heating of the higher power used in this proposed configuration with very brief stimulation with $H=1$ should be compared to the data in Figure 5 over the same overall time. Presumably, this will show rapid instantaneous heating that quickly dissipates and may provide further practical limitations of usHolo, possibly to require a sequential approach. Understanding these tradeoffs will be important for experimenters to decide on parameters in their experiments and to understand the use cases for cyclic-FLiT versus more conventional approaches.

10) Supplementary Note 2 should further cite and discuss limitations on usable laser power due to brain heating.

11) The new supplementary figures should be cited and discussed in the main text.

12) In the supplementary notes and tables the definitions of DT seem to be for only the first cycle of cyclic FLiT. While the cells will receive their first cycle of stimulation within such a short time, as I understand it, it would take $N_{cyc} * DT$ to get the overall stimulation time. The actual DT difference may be more relevant regarding spike time latency and jitter. This should be clarified.

13) For Supp Fig 21, please add the currents for conventional holography using the same powers shown for cyclic-FLiT. It may be also helpful to add graphs for the usHolo configuration.

14) Regarding the axial physiological PSF, the measurements in Supplementary Figure 15B suggest that the axial resolution suffers for the highest power $\sim 74\text{mW}$. This is similar to the power used in Figure 4 for FLiT. The remaining panels in Supplementary Figure 15 do not include the higher power measurements. The authors made several statements regarding the axial confinement being hindered by increased power, but use high power for the cyclic-FLiT experiments. For axial measurements, please indicate or include values that correspond to the parameters used for experiments throughout the paper.

Minor comments

1) The reference to Supplementary Figure 18 appears to be outdated and should be removed from Supplementary Note 2.

2) In the Supp Note 3, Config 2, there seems to be a repeated sentence regarding \sqrt{H} , and likely a missing statement about \sqrt{H} from the rebuttal.

3) Some text appears to be cut off by a graph in Supplementary Figure 22D.

4) In Supp Fig 22A-C, what is the photocurrent normalized to? Also, please define t_ill

REVIEWERS' COMMENTS

Reviewer #3 (Remarks to the Author):

The authors have made significant improvements to the manuscript regarding the details of their protocols and how they relate to stimulation efficiency. It is very helpful to have the additional descriptions of the experiment configurations, as well as the additional modeling exploring the parameter space. I believe this will be a great service to readers to better understand the tradeoffs. With the configurations laid out, it appears that there are important gaps in the exploration of this parameter space, and that key parameters are allowed to vary favorably for cyclic-FliT but not conventional holography. I have attempted to lay out my understanding of the different effects and what the relative difference in efficiencies may more accurately be between conventional holography and cyclic-FliT, below.

Fundamentally, I believe that two methods must be compared to one another using the same experimental parameters whenever possible. In the two main configurations described by the authors in their modeling work, most parameters are held constant but one parameter—time (Configuration 1) or laser power (Configuration 2)—is allowed to increase for cyclic-FliT but not for conventional holography. The authors then take the resulting increase in number of targetable neurons in the cyclic-FliT case to reflect a total improvement in efficiency without accounting for the impact of longer experiment time or increased laser power. In my last review, I had requested that the authors provide modeling and experiments for how much power would be required for conventional holography to match the number of targetable neurons with cyclic-FliT when total experiment time was held constant, or when total dwell time was held constant. In their rebuttal, the authors made some comments about the latter and did not explore the former. When the parameters are matched between methods, or when conventional holography is also permitted to vary parameters for its benefit, I believe that the differences between conventional holography and cyclic-FliT will be found to be far less than what the authors claim throughout the manuscript. This is fundamentally due to the inherent two photon absorption effect governing both methods.

If the relative efficiencies of conventional holography and cyclic FliT are different than claimed by the authors, as I believe they may be, the manuscript must be correspondingly revised. Even if the relative differences are not as pronounced when explored theoretically, there are potential applications for the cyclic-FliT, such as when a strict power ceiling is placed on experiments due to available laser power or damage thresholds. The authors touch on these factors, but the arguments are a bit two-sided and disjointed—arguing there are such limitations but allowing increasing power for cyclic-FliT to relatively high levels per cell and in a supplementary note implying that many more neurons could be targeted with even more power. More tightly describing the power ceiling in the context of practical and biological limitations might point to utility of cyclic-FliT. I also still believe that the hybrid-FliT approach has utility and continue to encourage the authors to focus more on elaborating this approach.

Major comments:

1) Configuration 1 – I do not think it is correct to say that the number of targeted cells is “H times” higher for cyclic FliT in Configuration 1. This statement is off by potentially a large margin once the overall experiment time is taken into account. In Config 1, the cyclic FliT is allowed to increase the experiment time by H fold, but this is not applied to conventional holography in the comparison. An experimenter could easily sequence through multiple holograms using conventional holography over the same longer experiment time, as suggested by the authors, but this is not formally explored. In this case, conventional holography would indeed be limited by the t_{SLM} . The difference in number of targeted neurons will be reduced in sequenced conventional holography by the off duty cycle of the SLM. Since t_{dw} is reported to be 5-10 ms, this off duty cycle (assuming $t_{SLM}=3ms$) would be $3/8$ (37.5%) or $3/13$ (23%). Cyclic-FliT in Config 1 would be theoretically able to fill this fractional off duty cycle time, so would be able to target 1.3 to 1.6 times more neurons in this example $[H/(H*t_{dw})/(tdw+t_{SLM})]$, or in simpler form: $1+t_{SLM}/tdw$. This is noteworthy, but much lower than the “H times” more cells stated by the authors. For example, for $H=23$, this is overstating the difference by over 14 times. In practice, achieving ~1.6 times more cells could be achieved within the same dwell time by increasing power by an estimated ~1.36 times, incorporating the quadratic relationship of power and excitation. This would likely be a much simpler adjustment for many researchers than implementing cyclic-FliT.

2) From this, I would argue that in Supplementary Table 1, the Maximum number of cells is: Conventional Holo: $N_{std_max} = (H*(tdw)/(tdw+t_{SLM}))* (P_{tot}/P')$

1)+2): The expression given by the referee at point 2) is correct, although the case described by the equation has the same drawback as the case of conventional holography using a total time t_{dw} and already discussed in the Table 1 and 2. That is, it lacks the possibility for simultaneous cell activation, which is the reason why we have proposed cyclic-FliT.

Indeed as expected by the referee, using the expression $N_{std_max} = (H*(tdw)/(tdw+t_{SLM}))* (P_{tot}/P')$ and the numbers given in the manuscript ($t_{dw} \sim 5-10$ ms, $t_{SLM} \sim 3ms$, and $H=23$), for conventional holography one could sequentially scan ~14-17 holograms (and so excite 14-17 times more cells or groups of cell than with a single hologram), while FliT will cyclically scan ~23 hologram (and so excite 23 times more cells (or group of cell) than using a single hologram) .

The aspect on which the referee still seems to be confused is the difference among sequential hologram projection and cyclic hologram illumination and the implication of these two configuration in the simultaneity of multitarget excitation.

Precisely, if we come back to the configuration proposed by the referee (i.e. sequential holography in a time equal to $tdw*H$), the delay (DT) between the spiking time of the first and last cell (or group of cells) will be equal to $DT=((H-1)*tdw=70-160ms$ (for $tdw =5-10ms$) while it will be $DT=(H-1)*t_{cyc} = \sim 1.10ms$ for cyclic-FliT.

Therefore, we confirm what stated in the manuscript that for simultaneous activation of multiple cells, the FliT approach enables a gain of H (23) times in the number of reachable cells using Configuration 1).

3) Configuration 2 – This configuration, where total experiment time is held constant, must be explored in the modeling using parameters where power is allowed to increase for conventional

holography. The authors claim that they can target \sqrt{H} more neurons in cyclic-FliT but use \sqrt{H} more power. They do not increase power for conventional holography by the same amount and ask how many more neurons can be targeted. Given a linear relationship between number of targeted neurons and power in conventional holography (for a fixed dwell time), one could target \sqrt{H} more neurons with \sqrt{H} more power. Indeed, looking at “Maximum Number of Cells” in Configuration 2 Supplementary Table 2, and replacing P_{cyc} with $P_{std} \cdot \sqrt{H}$, as defined in the first row, it appears the \sqrt{H} would cancel out the \sqrt{H} increase for Max number of cells in cyclic-FliT, wouldn't it? This implies that two methods are equivalent when one accounts for the increase in power. If more power (than \sqrt{H}) is required for conventional holography to target \sqrt{H} more neurons in the same total experiment time, how much more is required and why?

3) We realized that in Table 2 we have made a typo error in the expression for value of the power/cell needed for FliT this is $N_{max}^{cyc} = P_{tot}/P_{std} \cdot \sqrt{H}$ and not $N_{max}^{cyc} = P_{tot}/P_{cyc} \cdot \sqrt{H}$, which could explain the confusion of the referee.

As correctly stated elsewhere in the manuscript, figure captions and supplementary information: keeping the same spiking properties for FliT and conventional holography using the Configuration 2) requires raising the power/cell by \sqrt{H} in the case of FliT but also enables reaching H more targets, this corresponds to a net gain of \sqrt{H} times more cells.

We confirm what previously stated and experimentally demonstrated: given a total laser power P_{tot} , the net gain of reachable cells with FliT in Configuration 2 is of \sqrt{H} times.

4) For Figure 5, what is the total illumination time for cyclic illumination of a single neuron (How many cycles are repeated for each neuron in cyclic illumination, with what dwell time per cycle?)?

As stated in the manuscript at line 364, the characteristic of the cyclic illumination described in Fig.5 are: 10 cycles; 50 μ s illumination pulses; 20 holograms; 50 μ s*20=1ms per each cycle. We now added this information also in the caption.

If that total illumination time per cell in cyclic FliT is used for conventional holo, and power is increased to $\sqrt{H} \cdot P_{std}$ for the conventional holo case, what does the modeled heating look like? In other words, the nonlinear increase in two photon excitation with higher power is a major factor in reducing heating effects, but this same potential nonlinear benefit is not applied to the conventional holography case for fair comparison.

The temperature rise requested by the reviewer could directly be extracted from Fig.5D.

More in general, one should note that the temperature rise is linear with the illumination power.

Thus, if one increases the power to $P' = P_{std} \cdot \sqrt{H}$, the temperature rise will increase to $\Delta T' = \Delta T_{std} \cdot \sqrt{H}$.

Specifically, if for an illumination dwell-time $t_{dw} = 50\mu\text{s} \cdot 10 \text{ cycles} = 500\mu\text{s}$, the temperature rise under steady illumination is $\Delta T_{std} = 0.31\text{K}$, increasing the power of a factor $\sqrt{H} = \sqrt{20}$, will generate a temperature rise of $0.31 \cdot \sqrt{20} = 1.38\text{K}$.

To facilitate the visualization of these values, we now added a new Supplementary Figure 18.

5) The authors argue that increasing power for conventional holography would lead to damage. Indeed, increasing power will lead to more potential damage. However, this also applies to increasing power in the cyclic-FliT case, as in the case of using \sqrt{H} more power, with some discussion from the authors in Supplementary Note 2. My understanding is that the experiments typically used $H \sim 20$. $\sqrt{20} = 4.47$. That is a large increase in power. The authors argue that that amount of increase in power is OK for cyclic-FliT but that it would be damaging to increase power for conventional holography to attempt to match the same number of targeted neurons. The two must be compared on equivalent terms.

6) “However, because of the drastic nonlinear photodamage and thermal effect, this last configuration ($P_{std} \gg P_{sat}$) is very unlikely to be adopted in two-photon optogenetics.” This case/assumption should be explored and presented systematically in the text and figures. In the case of cyclic-FliT, the authors argue that they can use \sqrt{H} more power – e.g., which can be ~ 4.8 times when using 23 holograms, but they do not explore more power for conventional approaches. This analysis should incorporate nonlinear damage and instantaneous (during illumination) and average (over the same overall experiment time) thermal effects for both methods.

5)+6)

The referee here is confused among power/cell and total power. Indeed, the power/cell used in FliT is higher of \sqrt{H} than what used in conventional holography, but the total power used to reach N target is \sqrt{H} lower.

Anyway, both non-linear and linear damage effects have been already largely discussed in the manuscript, we therefore do not understand the requirement of the referee.

Briefly, in the case of *cyclic*-FliT, we demonstrated that to reach the same spiking properties as in conventional holography, *cyclic*-FliT requires configuration 2, that is \sqrt{H} more power PER CELL which indeed correspond to $\sqrt{H} \sim 4.8$ times when using 23 holograms. We then measured the photodamage effects (end of section ‘*Cyclic*-FLiT for multi-target excitation’) that such increased power PER CELL could generate and concluded that FliT can safely use up to a hundred of holograms before exceeding the non-linear photodamage threshold (see also Supplementary Notes 2). In the same section we also analyzed the nonlinear photodamage effect for conventional holography.

The thermal effects related to the average energy sent into the sample during the total time of the experiments have been also analyzed and discussed for FliT and conventional holography in a broad range of powers (please see also point #4, 8,9).

7) The authors state in the response to one of my questions that “As this scheme relies on a single hologram, and all targets are illuminated in parallel, the number of achievable cells with usHolo will be proportionally reduced”. I disagree with this statement. In fact, the number of achievable cells will be equal to the number of cells in the hologram and not reduced. I believe the power used to achieve comparable numbers of neurons as the cyclic-FliT case (which was more my question) would be the same \sqrt{H} factor, as outlined above.

Actually the number of achievable cells is defined by the number of spots addressable by the hologram AND the available laser power. In a fully parallel illumination approach, for a max

total power equals to P_{tot} and a chosen power per cell P' , the maximum number of cells that can be simultaneously stimulated with a single hologram is $N_{max} = P_{tot}/P'$.

This number will obviously be larger the lower is the power P' chosen per cell.:

e.g. if with $P_{tot}=1W$ and $P'=10mW$ one could reach $N_{max}=100$, while for $P_{tot}=1W$ and $P'=50mW$ one could only reach $N_{max}=20$

The configuration usHolo, suggested by the referee in the past report, minimizes the illumination time and therefore need to increase P' and thus reaches less cells with respect to a conventional configuration where a longer illumination time enables to work a lower value of P' . This is totally unrelated to how many spots a single hologram could generate.

We therefore confirm what previously stated: for a given total available power P_{tot} , with usHolo (high power, P' , per cell and short illumination time) the number of simultaneously achievable cells with a single hologram will be reduced with respect to a configuration using lower value for P' and longer illumination time.

8) The authors then state that the “maximum number of activatable cells” is proportionally lower with the increase in power, assuming a total power threshold. While technically true, the authors separately argue that several watts of power are available at the objective and also allow cyclic-FliT to use significantly more power than conventional holo (i.e., \sqrt{H} more power). This use of increased power for cyclic-FliT will also have limitations regarding the total available power that can be safely used in the experiment.

Here the referee still seems to be confused about the two main reasons for which we have proposed cyclic-FliT:

1) limited laser exit power:

In this case, given a maximum available laser power P_{tot} , conventional holography enables to reach $N = P_{tot}/P'$ cells; FliT enable to reach using THE SAME TOTAL POWER: \sqrt{H} to H more targets. Opposite to the referee statement, cyclic-FliT DOES NOT use significantly more total power than conventional holography, but it uses the same. Therefore the two approaches have the same limitations.

2) reduced thermal damages:

For extremely powerful lasers (i.e. for which the max number of cells is not limited by the available laser power), reaching a number N of cells in conventional holography will generate a sample heating proportional to $N \cdot P/P'$, in FliT this will be proportional to $N \cdot (P/(\sqrt{H} P'))$. Therefore, opposite to referee expectations, in this situation FliT enables reaching the same number of cells than conventional holography under safer conditions and not the other way around. (please also look at the section “Cyclic-FliT illumination enables to minimize thermal effects” and Figure 5)

9) In the rebuttal point “D”, Figure R8 does not really address my question, and the limitation of total power applied by the authors may not always be relevant in what they have termed “usHolo”. Furthermore, sequential “usHolo” could be implemented to work around total power limitations. How much power would be required when using conventional holography ($H=1$) and the same overall dwell time to stimulate as many neurons as cyclic-FliT (that uses \sqrt{H} more power for H holograms)? I think this would be $power_per_cell_std \cdot \sqrt{H} \cdot num_cells$. For cyclic-FliT, I believe it is $power_per_cell_std \cdot \sqrt{H} \cdot num_cells / H$ indicating the potential advantage. However, in the case of “usHolo”, the total illumination would be much briefer: t_std_dw / H compared with overall illumination time in the cyclic case being t_std_dw ,

a factor of H times longer. Taken over the same overall time t_{std_dw} the average power would be equivalent. Alternatively, the usHolo approach would permit another hologram to be generated in t_{SLM} and another set of neurons be targeted potentially sooner than with cyclic-FliT for $(t_{std_dw}/H+t_{SLM}) < t_{std_dw}$, leading to more overall neurons targeted. The usHolo configuration should be explored and discussed further by the authors and the effects on heating of the higher power used in this proposed configuration with very brief stimulation with $H=1$ should be compared to the data in Figure 5 over the same overall time. Presumably, this will show rapid instantaneous heating that quickly dissipates and may provide further practical limitations of usHolo, possibly to require a sequential approach. Understanding these tradeoffs will be important for experimenters to decide on parameters in their experiments and to understand the use cases for cyclic-FliT versus more conventional approaches.

See response to # 1-2, 4,5,6

10) Supplementary Note 2 should further cite and discuss limitations on usable laser power due to brain heating.

The manuscript includes an entire section dedicated to this issue (see also response to point #8).

11) The new supplementary figures should be cited and discussed in the main text.

The new supplementary figures are part of the new supplementary notes and therefore cited in the corresponding notes.

12) In the supplementary notes and tables the definitions of DT seem to be for only the first cycle of cyclic FliT. While the cells will receive their first cycle of stimulation within such a short time, as I understand it, it would take $N_{cyc} * DT$ to get the overall stimulation time. The actual DT difference may be more relevant regarding spike time latency and jitter. This should be clarified.

Here the referee seems to be confused about the definition of DT: this is the expected shift delay between the spiking time of the first and last excited cell (or group of cells) using cyclic-FliT or sequential conventional holography and therefore it is not related to the first cycle only. Also it has no effect on the cell jitter or latency.

Few clarifications:

For Configuration 2), $DT=(H-1) t_{cyc} \sim 1.15ms$ while for conventional holography reaching the same number of cells (or group of cells) requires sequentially refreshing among \sqrt{H} holograms, so that $DT= (tdwell+t_{SLM})x(\sqrt{H}-1) \sim$ hundred of millisecond (please look at Supplementary Table 2).

The spike latency and jittering reachable with the two approaches is the same as explained in the manuscript (section 'Cyclic-FLiT for multi target excitation') and demonstrated in Figure 3.

For configuration 1) $DT=(H-1) t_{cyc} \sim 1.15 ms$, while for conventional holography reaching the same number of cells (or group of cells) requires sequentially refreshing among H holograms, so that $DT= (tdwell+t_{SLM})x(H-1) \sim$ hundred of millisecond (please look at supplementary Table 1).

In this case, it is the longer current rise time, and not DT, for cyclic FLiT that generates longer latency and jittering, as already discussed and demonstrated in the manuscript (Supplementary Figure 10).

13) For Supp Fig 21, please add the currents for conventional holography using the same powers shown for cyclic-FLiT. It may be also helpful to add graphs for the usHolo configuration.

Figure supplementary 21 has been prepared to explain why, in the conditions of configuration 2, cyclic-Flit reaches the same current as conventional holography using \sqrt{H} less power. The comparison that the referee is asking to add makes no sense to us.

14) Regarding the axial physiological PSF, the measurements in Supplementary Figure 15B suggest that the axial resolution suffers for the highest power $\sim 74\text{mW}$. This is similar to the power used in Figure 4 for FLiT. The remaining panels in Supplementary Figure 15 do not include the higher power measurements

It actually does include it: the last point in panel C, corresponding to the broadest curve of panel B, represent the power $16\sqrt{H}$ which is also the higher power shown in panel D and E.*

The authors made several statements regarding the axial confinement being hindered by increased power, but use high power for the cyclic-FLiT experiments. For axial measurements, please indicate or include values that correspond to the parameters used for experiments throughout the paper.

All-optical experiments of Figure 4 have been performed to demonstrated that FLiT (using configuration 2) enables reaching \sqrt{H} times more cells than conventional holography. We aimed at demonstrating this gain on a large power range (from 5 to 33) \sqrt{H} mW). To be noted, calcium transients and APs are already reliably generated in the low power range (Fig.4f and Suppl. Fig.15d).*

*Single-cell electrophysiological experiments in Supplementary figure 15 have been done to demonstrate that FLiT despite using $\sqrt{H}*P'$ power/cell reaches the same axial resolution than conventional holography using P' . An effect also discussed in supplementary Figure 21. We performed this in a range of power enough to demonstrate the effect.*

The conclusions derived in the two experiments are unrelated and, as such, we do not see why these two experiments had to be performed at the same powers.

Minor comments

1) The reference to Supplementary Figure 18 appears to be outdated and should be removed from Supplementary Note 2.

Done

2) In the Supp Note 3, Config 2, there seems to be a repeated sentence regarding \sqrt{H} , and likely a missing statement about \sqrt{H} from the rebuttal.

Done

3) Some text appears to be cut off by a graph in Supplementary Figure 22D.

Modified.

4) In Supp Fig 22A-C, what is the photocurrent normalized to? Also, please define t_{ill}
Photocurrents are normalized to the maximal achievable photocurrent. We removed t_{ill} as it corresponds now to t_{dw} defined in the caption.